**The importance of blowing snow to halogen-containing aerosol in coastal Antarctica: Influence of source region versus wind speed**

Michael R. Giordano[1], Lars E. Kalnajs[3], J. Douglas Goetz[1,a], Anita M. Avery[1,b], Erin Katz[2], Nathaniel W. May[4], Anna Leemon[4], Claire Mattson[4], Kerri A. Pratt[4], Peter F. DeCarlo[1,2]

[1]Department of Civil, Architectural, and Environmental Engineering, Drexel University, Philadelphia, Pennsylvania, USA
[2]Department of Chemistry, Drexel University, Philadelphia, Pennsylvania, USA
[3]Laboratory for Atmospheric and Space Physics, University of Colorado at Boulder, Boulder, Colorado, USA
[4]Department of Chemistry, University of Michigan, Ann Arbor, Michigan, USA
[a]Now at Laboratory for Atmospheric and Space Physics, University of Colorado at Boulder, Boulder, Colorado, USA
[b]Now at Aerodyne Research, Inc., Billerica, Massachusetts, USA

*Correspondence to*: Peter F. DeCarlo (pfd33@drexel.edu)

**Abstract.** A fundamental understanding of the processes that control Antarctic aerosols is necessary in determining the aerosol impacts on climate-relevant processes from Antarctic ice cores to clouds. The first in situ observational online composition measurements by an aerosol mass spectrometer (AMS) of Antarctic aerosols were only recently performed during the 2-Season Ozone Depletion and Interaction with Aerosols Campaign (2ODIAC) field campaign. 2ODIAC was deployed to sea ice on the Ross Sea near McMurdo Station over two field seasons: Austral spring-summer 2014 and winter-spring 2015. The results presented here focus on the overall trends in aerosol composition primarily as functions of air masses and local meteorological conditions. The results suggest that the impact of long range air mass back trajectories on either the absolute or relative concentrations of the aerosol constituents measured by (and inferred from) an AMS at a coastal location is small relative to the impact of local meteorology. However, when the data is parsed by wind speed, two observations become clear. First, a critical wind speed is required to loft snow from the surface, which, in turn, increases particle counts in all measured size bins. Second, elevated wind speeds showed increased aerosol chloride and sodium. Further inspection of the AMS data shows that the increased chloride concentrations have a more "fast vaporizing" nature than chloride measured at low wind speed. Also presented are the Cl:Na ratios of snow samples and aerosol filter samples, as measured by ion chromatography, as well as non-chloride aerosol constituents measured by the AMS. Additionally, submicron aerosol iodine and bromine concentrations as functions of wind speed are also presented. The results presented here suggest that aerosol composition in coastal Antarctica is a strong function of wind speed and that the mechanisms determining aerosol composition are likely linked to blowing snow.

## 1 Introduction

Despite the logistical difficulties in performing atmospheric field work in Antarctica, measurements of tropospheric aerosols over and around the continent have been conducted for nearly half a century. Direct, online measurements by Voskresenskii (1968) and Warburton (Warburton, 1973; Hogan, 1975) were the first studies to demonstrate, over multiple

sites on the continent, that total number concentrations of Antarctic aerosols appear to be highly dependent on air mass origins (marine vs. continental). In conjunction with these early online concentration measurements, offline aerosol measurements were being simultaneously conducted at various points along coastal and inland Antarctica. These offline measurements, conducted primarily with impactors and x-ray diffraction measurements, suggested that the Antarctic aerosol population was

primarily between 200 nm to 2 μm and largely, by mass, composed of sea salts and sulfates in various forms (Cadle et al., 1968; Fischer et al., 1969). Cadle et al. (1968) postulated that the source of Antarctic sulfate originated in the stratosphere, though it has been recently shown that sulfates over the continent are most likely not stratospheric in origin and are due to emissions of Southern Ocean biota and an unknown source inside the continent (Legrand and Wagenbach, 1998; Giordano et al., 2017). After these early studies, longer term campaigns at continuously occupied McMurdo and South Pole Stations

determined both number concentrations and composition of aerosols in continental Antarctica (e.g. Cobb, 1973; Gladney et al., 1972; Duce et al., 1973; Ondov et al., 1973). These long-term projects confirmed that sea-salt and sulfates dominate the chemical composition of Antarctic aerosols with total PM mass generally always below 5 μg m$^{-3}$. Aerosol number concentrations were generally consistent with the few previous studies, despite the fact that contamination of samples from local point sources was (and continues to be) a source of uncertainty on the continent (e.g. Gladney et al., 1972; Rosman et al.,

1994). These early Antarctic aerosol studies, though limited by instrument sensitivities and detection limits, provided the preliminary observational data sets on aerosol concentration and composition which have guided much of the research since.

After about 1975, Antarctic tropospheric aerosol research shifted from primarily exploratory studies towards focused investigations of aerosol transport, sources, and sinks over the continent. This paradigm shift was likely brought about due to increased focus on Antarctic atmospheric science and the development of increasingly sensitive aerosol instrumentation (Shaw,

1988). The early aerosol number concentration measurements were expanded on greatly in this time period due to this burgeoning interest in aerosol sources and transport. One of the most important findings was the discovery that nucleation mode ($d_p < 100$ nm) concentrations, at both the Pole and along the coast, exhibit strong seasonality in terms of concentrations (e.g. Hogan and Barnard, 1978; Ito, 1982; Jaenicke and Stingl, 1984; Gras and Andriaansen, 1985; Ito, 1985; Shaw, 1986; Parungo et al., 1989). This seasonality of nucleation particles has been observed consistently since continuous concentration

records have been kept on the continent (e.g. Rankin and Wolff, 2003). Most of this seasonality has since been linked to oceanic dimethyl sulphide (DMS) emissions as Southern Ocean phytoplankton activity cycles between a minimum in the Austral winter to a maximum in the summer (e.g. Shaw, 1979; Wagenbach et al., 1988; Preunkert et al., 2007; Giordano et al., 2017). In addition to the seasonality of nucleation mode particles, the interest in Antarctic aerosol sources contributed to the determination of the chemical composition of aerosols of all sizes. Similar to previous measurements, sulfate and sea salt

particle types were found to dominate the total aerosol number population. In terms of aerosol size, sulfates are predominantly found in the nucleation and accumulation modes, whereas sea salt particles dominate the coarse mode and accumulation mode mass. (e.g. Wagenbach et al., 1988; Parungo et al., 1989; Wouters et al., 1990; Rankin and Wolff, 2003). Though there is a suggestion of multiple size modes for reported aerosol components, most measurements have reported unimodal size distributions (dating back to, e.g., Ito, 1985). This unimodality, and the fact that aerosol from the interior of the continent

contain sulfates, black carbon, and crustal materials, has been taken to indicate that long-range transport is a major factor in aerosol physical and chemical properties over the continent (e.g. Maenhaut et al., 1979; Hansen, 1988; Ito, 1989). Ammonium and various nitrogenated compounds have also been identified as minor components of Antarctic aerosols and have been linked to various Antarctic fauna (Legrand et al., 1998).

5       The modern era of Antarctic aerosol science has focused on bringing to bear instrumentation of ever-increasing sensitivities and resolution to examine the impact of sea, air, snow and ice on tropospheric aerosol. The range of questions that have been examined is broad. Subjects ranging from the deposition of aerosols to snow (Wolff et al., 1998), the extent of the impact of oceanic sulphur to Antarctic aerosols (Prospero et al., 1991; Minikin et al., 1998; Wagenbach et al., 1998a), to the depletion of ions from sea-salt derived aerosols (Jourdain and Legrand, 2002; Jourdain et al., 2008), to the presence of trace

marine elements (Li, K, Mg, Ca, Sr) in coastal Antarctic aerosols (Weller et al., 2008) have all been examined. Though this era has seen new instrumentation brought to the continent to study Antarctic aerosols, online high-resolution aerosol composition measurements have still yet to be taken. The closest measurements come from ship- or island-based measurements in the Southern Ocean that used aerosol mass spectrometers (Zorn et al., 2008; Schmale et al., 2013) as well as the deployment of an ambient ion monitor-ion chromatography (AIM-IC) system during the OPALE campaign (Pruenkert et al., 2012). The

ability to measure aerosol composition online and at high time resolution allows for questions regarding transient phenomena to be investigated. Answering questions such as the sources and impact of wind speed on Antarctic aerosols are necessary to further our understanding of background aerosol processes and therefore our ability to confidently reconstruct paleoclimate records from ice cores. This gap in the science was the primary driving force behind the 2ODIAC campaign, which took place over two field seasons in 2014 and 2015 on the sea ice near McMurdo station. The deployment of an aerosol mass spectrometer,

and an accompanying suite of gas-phase and aerosol size and number instruments, has shown that aerosol-phase sodium and chloride are enhanced during blowing snow/high wind speed events and that air-mass back trajectories have little impact on the overall aerosol composition as measured at a coastal site.

      Halogen-containing aerosols in the Antarctic are connected to the depletion of near-surface tropospheric ozone, a greenhouse gas, (Kalnajs et al., 2013) and may also contribute to new particle formation (Sipilä et al., 2016). Numerous remote

sensing measurements of halogen oxides in the boundary layer (Saiz-Lopez et al., 2007; Simpson et al., 2015; Prados-Roman et al., 2018) show active bromine and iodine chemistry in the Antarctic troposphere. Chemical ionization mass spectrometry measurements of $Br_2$, BrCl, and BrO confirmed the coupling of bromine and chlorine chemistry in the springtime Antarctic troposphere (Buys et al., 2013). Reactions with non-methane hydrocarbons (e.g. Ramacher et al., 1999), elemental mercury (Ariya et al., 2004; Saiz-Lopez et al., 2008), and DMS (von Glasow et al., 2004; Read et al., 2008) can all have large impacts

on tropospheric composition over the poles. Additionally, some work has also been done examining iodine oxides' role in forming ultrafine aerosols (O'Dowd et al., 1998; McFiggans et al., 2004; Saunders and Plane, 2005; Sipilä et al., 2016). Despite significant progress in our understanding of atmospheric halogen gases, a lack of observational data capturing the interactions between gas-phase halogens, aerosols, and snow persists, limiting our understanding of these multiphase processes.

Over remote and maritime regions, meteorology and wind speed in particular has long been known to play a major role in both aerosol generation and controlling aerosol properties. Properties ranging from aerosol composition, to size distribution, to optical properties are modified through the lofting of crustal materials and wave driven sea salt aerosol production (e.g. Monahan, 1968; O'Dowd and Smith, 1993; Smirnov et al., 2003; Kandler et al., 2008; Gantt et al., 2011). How Antarctic aerosols (and potentially cryospheric aerosols in general) are impacted by meteorology is not well understood. Some studies in Antarctica have hypothesized that elevated wind speeds can cause sea salt concentration differences through unknown mechanisms, but previous low temporal resolution measurements (or other instrument limitations) diminish the ability of such studies to draw definitive conclusions (e.g. Wagenbach et al., 1998; Hara et al., 2004). Proposed mechanisms for enhanced sea salt aerosol concentrations include sublimating saline snow hydrometeors which is hypothesized to produce sea salt aerosol at an order of magnitude higher production rate than over open ocean (Yang et al., 2008). This 'blowing snow' mechanism has been implemented in global models and various implementations have seen varying success in simulating sea salt concentrations (Levine et al., 2014; Huang and Jaeglé, 2016; Rhodes et al., 2017). However, there is the lack of experimental studies investigating the proposed mechanism of sea salt aerosol production from blowing snow, and the experimentally measured aerosol production rates. The results presented here, which vary in temporal resolution from low (days) to very high (minutes), suggest that coastal Antarctic aerosol composition cannot be explained without taking local meteorology (e.g. wind speed and direction) into account. Though the results presented here do not identify specific mechanisms, they strongly suggest that blowing snow may drive the composition dependence and aerosol enhancement observed as a function of wind speed, and further studies are necessary to investigate both production rates and mechanisms.

## 2 Methods

### 2.1 Measurement Periods and Field Sites

The 2ODIAC campaign set up measurement locations on coastal Antarctica near, but outside the influence of, McMurdo Station. Field site locations, including maps, for both periods of measurements are presented in detail in Giordano et al. (2017).

One of the main goals of the 2ODIAC campaign was to characterize the composition of Antarctic aerosols and track changes in composition as a function of seasonal changes in daylight (i.e. the transition from perpetual darkness, to twilight, to perpetual daylight as the Austral winter transitions to spring, summer). To accomplish this goal, the campaign was composed of two periods of intensive measurements: the first from the late Austral spring to mid-summer (28 October- 03 December) of 2014 and the second from late Austral winter to early spring of 2015 (08 September – 12 October). The 2014 spring/summer campaign period began measurements shortly after 24-hour daylight days commenced (24 October 2014). The 2015 winter/spring campaign began measurements shortly after the Antarctic sunrise and continued as full daylight hours increased from approximately 9 hours of twilight per day to 17 hours of sun per day. SI Figure S1 visualizes the hours of daylight and solar angle at noon measured at the field site for both field seasons.

## 2.2 Instrumentation

The instrumentation deployed during the 2ODIAC seasons can be broken into two primary categories: gas-phase instruments and aerosol-phase instruments. Both the gas- and aerosol- phase instruments were housed and operated in the instrumentation hut. The gas-phase instrumentation suite - an ozone monitor (Thermo Environmental model 49C) and a $NO_x$ monitor (Thermo Environmental model 42C) in both field seasons - was operated on separate Teflon® inlet lines that were mounted 1m above the instrumentation hut. The aerosol-phase instrumentation suite was comprised of both online and off-line instruments to measure a wide range of aerosol properties. In both field seasons aerosol composition was measured by a Soot Particle Aerosol Mass Spectrometer (SP-AMS; Aerodyne Research Inc., Billerica MA; Onasch et al., 2012). The SP-AMS consists of an Aerodyne High-Resolution Time-of-Flight Aerosol Mass Spectrometer (HR-ToF-AMS; DeCarlo et al., 2006) combined with a soot vaporizing laser (from Droplet Measurement Tech., Boulder CO), though the soot laser was not used during 2ODIAC. Transmission efficiencies for different instruments vary, but generally particles between 60nm - 600nm vacuum aerodynamic diameter are transmitted with unit efficiency, with fraction of transmitted particle decreasing for larger sizes (Liu et al. 2007). Only particle bulk composition is discussed here due to the lower signal/noise associated with particle sizing. Total particle number concentration was also measured in both field seasons by a water-based condensation particle counter (7 nm - 3 µm; 3783 Environmental Particle Counter; EPC, TSI). Aerosol size distribution measurements were accomplished through two different suites of instruments, one used in 2014 and the other in 2015, due to instrument availability issues but both field seasons characterized aerosol size and number distributions over similar ranges. In 2014, aerosol sizing was performed by an Ultra-High Sensitivity Aerosol Spectrometer (~55-1000 nm; UHSAS, Droplet Meas. Tech.) and a Scanning Electrical Mobility Spectrometer (~9-850nm; SEMS, Brechtel Man. Inc.). In 2015, aerosol sizing was performed by a Scanning Mobility Particle Sizer (~10-420nm; SMPS, TSI, 3080/3081 SMPS and 3787 CPC) and an Optical Particle Counter (0.3-25µm; Lighthouse Remote 3104). In addition to the online instruments, Teflon® aerosol filters were also collected during both field seasons. The filters were collected for approximately 24 hours each with a sample rate of 15 L min$^{-1}$. In both field seasons, the aerosol instruments sampled off of a stainless steel manifold connected to a stainless steel sampling inlet that went through the roof of the instrumentation hut. The inlet system was kept at sub-turbulent Reynolds' numbers to insure minimal loss of particles to the inlet walls. The inlet was shielded to prevent direct sampling of windblown snow and was heated to 30°C (as measured at the portion of the inlet inside the fish hut) to prevent riming. The inlet heating resulted in sampling dry aerosol (confirmed by a series of test in the field). Gas-particle partitioning due to the heated inlet is possible but could not be measured with available instrumentation and is assumed to be negligible for the major species of interest measured by the AMS. No cyclone was used but the inlet cover geometry prevented large particles/droplets/snow from being sampled. Using a particle loss calculator and the geometry of the aerosol inlet (von der Weiden et al., 2009), the overall transmission efficiency of the inlet was calculated to be >95% for particles <1µm, 50% at 5µm, and 0% for particles >9µm. In addition to inlet losses,

the standard AMS lens transmission shows reduced transmission of particles larger than ~600 nm in vacuum aerodynamic diameter (dva; Liu et al., 2007).

In both field seasons, consistent daily and weekly routines of instrument verification and calibration were followed. The daily routine consisted of a 1 hour period of a HEPA filter being applied to the aerosol inlet, at different times during the day. The 1 hour period provided a check that all aerosol counters read 0 counts and the inlet system was not leaking, allowed for the AMS to calculate a background signal, and provided time to ensure that various instrument performance metrics (e.g. laser voltages) were not indicating systemic problems. Aerosol filters were also switched during this time period. Once per week in-depth calibrations were performed. In the AMS, ionization efficiency calibrations were conducted with ammonium nitrate (selected at 350nm mobility diameter) and particle time-of-flight (PToF) size calibrations were conducted with polystyrene latex nanobeads (60, 100, 300, 600, and 800 nm). Both AMS parameters across both field seasons varied by less than 10%. For the particle counters (EPC, CPCs, UHSAS), ammonium nitrate was atomized into the SMPS or SEMS and the output of the particle counters compared against each other. Weekly zero checks were also conducted for the gas-phase instrumentation suite.

A co-located weather station (Davis Vantage Pro2) was used to record meteorological data for both field seasons, and an anemometer (rated 0-200 km h$^{-1}$), a temperature and relative humidity probe (0.5ºC and 1% accuracy, respectively), and solar radiation sensors (spectral response 400-1100nm, 1 Wm$^{-2}$ resolution, ±5% accuracy at full scale) made up the sensor suite that was mounted 2.5 m above ice level, 50 m NE of the sampling hut. Additionally, in 2015, a laser disdrometer (Parsivel 2, OTT Hydromet) was installed on the roof of the instrumentation hut. Due to the issues with non-vertical precipitation, the disdrometer does not provide a reliable precipitation size or speed record but instead provides a way to determine if and when blowing snow was present.

A collection system for wind-blown snow was also set up 10 m away from the meteorology sampling site. The collection system was set on the snow/ice surface and included four collection vials with a wind break to induce stagnation in the air stream: two set up to capture the northwesterly blown snow and two to capture the southeasterly blown snow. The snow samples were collected when there was adequate snow in 50mL polyethylene plastic centrifuge tubes for future ion chromatography analysis. During high wind speed periods, adequate collection generally occurred every 24-48 hours. During low wind speed periods, collection took much longer (4-7 days) due to a combination of lower airborne snow loading and increased sublimation of captured snow.

## 2.3 Quantifying Species of Varying Speeds of Thermal Desorption in the AMS

The AMS is generally considered a quantitative instrument for non-refractory, submicron (40nm – 1μm vacuum aerodynamic diameter) aerosol components such as sulfates, nitrates, organic species, and ammonium (e.g. Canagaratna et al., 2007 and references therein). For refractory aerosols such as crustal aerosols, however, the AMS is not able to vaporize the components of the aerosol, and consequently has no ability to measure these species. Some chemical species are semi-

refractory, such that they vaporize slowly from the tungsten heater in the AMS ionization region. The concentration of these species can be inferred from changes in the background signal in the AMS (e.g. Salcedo et al 2010). A full explanation and discussion about the operating principles of the AMS is outside the scope of this manuscript but the reader is referred to Jayne at al. (2000) and DeCarlo et al. (2006) for more details. In simplistic terms though, an aerosol particle enters the AMS and is vaporized on a heater. The heater is generally set at 600°C because that temperature has been identified as a balance between particle vaporization of a majority of aerosol components and minimizing fragmentation of aerosol components. The resulting vapor is then ionized with resulting ions measured by a high-resolution time-of-flight mass spectrometer. The measured mass spectra also contain signal from the instrument background and atmospheric gases. To subtract, the background mass spectra are also recorded when the aerosols are blocked from the vaporizer (and therefore the ionizer and spectrometer). When aerosols pass through to the vaporizer, the AMS is operating in "open" mode. When aerosols are blocked, the AMS is operating in "closed" mode. The AMS is generally run with multiple open-closed cycles being averaged into one mass spectra. An implicit assumption of this operating scheme is that aerosol-phase components vaporize quickly - i.e., aerosol-phase components desorb from the heater on a time scale much faster than the length of the "closed" operating mode. This assumption generally holds true for most organic, nitrate, sulfate, and ammonium aerosol species but does not hold for substances with high temperatures of vaporization such as salts, soot, etc. Though many salt species do partially vaporize in "open mode", these species are termed "slowly vaporizing" because they continue to vaporize off the heater in the "closed" mode. This continuing vaporization results in an elevated subtraction from the "open" mode and therefore an under reporting of concentrations. In non-coastal and anthropogenically dominated aerosol mass populations, this effect is generally negligible for species such as Cl for the aerosol size range measured by the AMS (Drewnick et al., 2015). However, where sea salts dominate the aerosol mass, such as in Antarctica, this effect cannot be ignored (Allan et al., 2004).

One way to correct for the slow vaporization of certain species is to determine the loss of mass of a semi-refractory species from vaporizer as a function of time. By determining this loss rate, the total initial aerosol mass can be calculated from a straightforward mass balance approach using the measured signal in the background of the AMS. Salcedo et al. (2010) used such a mass-balance approach to calculate the decay constant for slow vaporizing lead in the AMS. The model employed by Salcedo et al. (2010) assumes an incoming mass flux of a slow-vaporizing species which impacts the vaporizer, sticks to it, and then desorbs at a rate proportional to the amount on the vaporizer at any given time. This assumption results in an explicit ordinary differential equation. The same method has been adopted for chloride (measured in the AMS, Chl) and sodium (measured in the AMS, $Na_{AMS}$) in the analysis presented here. The first step in the process of calculating slow vaporizing species in the AMS is to determine the mass loading on the vaporizer. Salcedo et al. (2010), measuring lead, and Drewnick et al. (2015), measuring chloride salts, went about this process by performing controlled experiments in laboratory settings by atomizing solutions of known composition into the AMS. Atomizing a known solution is useful because it allows for precise control over the timing of slow-vaporizing aerosol in to the AMS. We also have performed laboratory atomization tests, but because exact instrumental conditions are different from the field, we chose to use fortuitous field data to calculate Chl decay

rates. In this case, a sudden and abrupt wind speed change which ended a period of elevated Chl (and total aerosol mass) is used to determine the decay rate of chloride on the AMS vaporizer.

We discuss the procedure applied for Chl here, but the analogous method with a different exponential decay constant was determined independently for $Na_{AMS}$. Figure 1 shows how the wind speed changes affect the closed signal of Chl in the

AMS. Previous to the wind speed change, incoming chloride concentrations as measured by the AMS were "high", i.e. Chl was building up on the vaporizer, which is seen in the elevated closed signal. After the wind speed drops, incoming Chl concentrations fall and the chloride built up on the vaporizer begins to burn off. This decay of the closed signal Chl can then be fit using an exponential decay model as described in Salcedo et al. (2010). The $\tau$ of the exponential decay model determined from this method is 40 min$^{-1}$ (see SI Fig. S2). An $e$-folding time of 40 minutes demonstrates why the daily 1 hour filter periods

for assessing AMS backgrounds (S2.1) are not sufficient for calculating a Chl decay rate. The results in this paper may therefore be slightly under- or over-estimated in terms of absolute concentrations aerosol of Na and Cl. Figure 1 also shows the reconstructed "total" signal from the measured (open – closed) and closed signal Chl. The total signal shows the sudden drop in Chl concentrations that would be expected with the wind shift dropping total aerosol mass. In this manuscript, this corrected total signal is used only in determining the "true" amount of slow vaporizing chloride, i.e. $m/z$ 35 and 37 ($^{35}Cl^+$ and $^{37}Cl^+$). This

correction method is not applied to fast vaporizing chloride-tungsten-oxide complexes discussed later in this manuscript for the reasons discussed in Drewnick et al. (2015).

In the analysis presented here, a relative ionization efficiency (relative to nitrate, $RIE_{NO3}$) of 1.3 is used for Chl. Laboratory tests indicate that the relative ionization efficiency of $Na_{AMS}$ in the Drexel AMS is 1.59 with respect to Chl, at least when measuring NaCl. Drewnick et al. (2015) suggests that RIE values for semi-refractory species may be heavily dependent

on what complexes are measured by the AMS (e.g. NaCl vs. $Na_2SO_4$ x $10H_2O$, etc.). Here this effect is ignored because of the inability to know a priori all of the complexes chloride and sodium may occur in the data. Therefore, the overall concentrations presented here may not be quantitative. However, qualitative comparisons in the data sets are still valid, especially in the forms presented here.

**2.4 Filter and Snow Analysis**

During the 2ODIAC campaign, aerosol filters and snow samples were collected to provide long integration time counterpoints to the AMS and other gas- and aerosol-phase instrumentation suite. After a filter or snow sample was collected, the samples were stored in a -20 °C freezer in the Crary Lab in Antarctica. At the conclusion of each campaign, samples were shipped frozen to Drexel University. All the samples were later sent frozen to the University of Michigan for analysis. Before analysis, the Teflon® aerosol filters were cut in half. One half was sonicated in nanopure water for 60 minutes and the other

half stored for future studies. The sonicated filter extract was then transferred to a sample syringe for IC analysis. The snow samples received no additional preparation apart from thawing and transferring to a sample syringe for IC analysis.

A Dionex ICS-1100 and ICS-2100 ion chromatographs were used to analyze the following inorganic ions in solution: $Na^+$, $NH_4^+$, $Ca^{2+}$, $Mg^{2+}$, $K^+$, $Cl^-$, $NO_2^-$, $SO_4^{2-}$, $Br^-$, and $NO_3^-$ (LODs: 2, 0.2, 0.2, 0.1, 0.04, 0.9, 0.3, 0.3, 0.09, and 0.3 µM,

respectively). For analysis of snow samples the ICS-1100 and ICS-2100 were each equipped with a 200 µL sample injection loop. For filter extract analysis the ICS-1100 and ICS-2100 were equipped with Ultralow Pressure Trace Concentrator Columns for Reagent-Free IC (ICS-1100: IonPac TCC-ULP1 RFIC, 5 x 23 mm, Dionex; ICS-2100: IonPac UTAC-ULP1 RFIC, 5 x 23 mm, Dionex) and exactly 1 mL was injected for each run. For all runs the ICS-1100 and ICS-2100 were each
equipped with a guard column (ICS-1100: AG18 RFIC, 4 x 50 mm, IonPac; ICS-2100: UTAC-ULP1 RFIC, 5 x 23 mm, Dionex), analytical column (ICS-1100: CS12A-5 µm, RFIC; ICS-2100: AS18 RFIC, 4 x 50 mm, IonPac), a suppressor (ICS-110: CERS-4 mm 500 RFIC, ICS-2100: AERS500 4 mm, Dionex), and a heated conductivity cell (DS6; Dionex). Methanesulfonic acid (20 mM) was used as eluent for the cation column and KOH gradient generated by an EGC III KOH system was used as eluent for the anion column. All samples were run in triplicate and were injected into the ICs through 0.22
µm PVDF filters.

## 3 Results and Discussion

A strong correlation was observed between the local wind speed and aerosol size, number and composition. The snow and 24-hour integrated filter results are presented first in section 3.1. Section 3.2 then examines the air mass back trajectories and local wind speeds of both field seasons to provide context for Sections 3.3, 3.4, and 3.5: aerosol number distribution
changes in terms of air mass back trajectories and wind speed, the AMS results analysed in terms of air mass back trajectories, and the AMS results analysed in terms of wind speed. Section 3.6 provides an examination of the chloride to sodium ratios measured by the AMS as a function of wind speed and direction. Section 3.7 presents a case study on aerosol iodine and bromine concentrations to examine halogens other than chloride in the aerosol phase. The first five subsections point to blowing snow being a major determinant of aerosol composition over coastal Antarctica while the last subsection examines the overall
halogen cycle over Antarctica.

In this manuscript, three notations are presented for chemical identities and correspond to either instrument used to quantify or to their presence regardless of molecular structure. 1) For any ion measured with the IC system, ionic charge is noted (e.g. $Cl^-$, $Na^+$). 2) For compounds measured with the AMS, notation in the text follows AMS standard notation (e.g. Chl for chloride, $SO_4$ for sulfate, $NO_3$ for nitrate) with the exception of sodium measured by the AMS ($Na_{AMS}$) and corrected
through the process described in S2.3. 3) For general and/or hypothetical discussions about concentrations without implication to measurement or molecular compounding, elements are referred to simply by their periodic table notation or written out (e.g. chloride, Cl, sodium, Na).

### 3.1 Integrated Aerosol Filter and Snow Measurements

Though the 2ODIAC campaign was focused on the first ever deployment of an AMS to Antarctica, examining the
aerosol filter and blowing snow data first provides insight into the supermicron aerosol and blowing snow that were not measured by the AMS. The filters captured both submicron and supermicron particles but the mass and ion loadings measured

on the filters are expected to be dominated by the latter. Since the AMS does not measure particles over 1 µm (vacuum aerodynamic) diameter, direct comparisons between the filter samples and AMS are not possible, but provide information about the relative importance of sulfate and other chemical species in the supermicron aerosols. Since both the filters and snow samples have a relatively low temporal resolution (24-hour minimum) relative to the variability in wind speed and direction, on average, it should be noted here that filter concentrations were typically higher in the 2015 campaign when wind speeds were consistently high.

The results presented here show two main points: one, that 2ODIAC measurements are consistent with previous studies results regarding seasonal differences in overall composition and ion ratios but, two, that some of the proposed mechanisms by which those seasonal differences arise are not supported by the observations from the 2ODIAC data and air mass back trajectory analysis.

### 3.1.1 Filter Samples

$Na^+$ and $Cl^-$ originating from ocean sources has long been known to make up the majority of the Antarctic aerosol mass (e.g. Wagenbach et al., 1998a; Rankin and Wolff, 2003). Both the filters and snow samples from 2ODIAC indicate that aerosol likely to be of oceanic origin also made up the majority (>80%) of the aerosol mass measured during the two seasons on the sea ice. Given the body of previous measurements on the continent (e.g. Rankin and Wolff, 2003) and the field site's location on annual sea ice (16 and 8 km from the ice edge for the 2014 and 2015 field seasons, respectively), this is not unanticipated. Figure 2 presents $Cl^-$ vs. $Na^+$ results from the filters taken during both Austral spring (2015) and summer (2014). Figure 2 shows that during springtime measured $Cl^-:Na^+$ ratios are at or slightly enhanced above seawater ion ratios, while during the summer, $Cl^-:Na^+$ is at or below seawater ratios. It should be noted that the measured summer levels are frequently below the limit of detection (0.944 µM $Cl^-$ or 2.353 µM $Na^+$), or higher in the blank filters used for corrections than the sampled filters, so this is not conclusive. Additionally, the total Cl and Na mass is generally ten times higher in spring than in summer. The increased spring mass (as compared to the summer mass) and the overall mass concentrations are consistent with the literature though it is well recognized that site-specific variables (e.g. topography) play major roles in overall sea salt aerosol mass (e.g. Hall and Wolff, 1998; Wagenbach et al., 1998a; Rankin et al., 2000; Jourdain and Legrand, 2002; Rankin and Wolff, 2003; Hara et al., 2004). In previous studies, the cause of this anti-correlation between sea salt aerosol concentration and distance to open water area was not clear. Virkkula et al. (2006) noted chloride losses (relative to $Na^+$) linked with increasing air mass residence time over continental Antarctica. Hall and Wolff (1998) observed that increased sea salt concentrations in the winter at Halley Base were linked to moderate (3-8 ms$^{-1}$) wind speeds. Hall and Wolff (1998) hypothesized that moderate wind speeds opened up polynyas in the sea ice which then quickly freeze and create a concentrated brine layer on top of newly formed ice. This brine layer then, with the possible aide of mirabilite ($Na_2SO_4 \cdot 10H_2O$) and frost flower formation, aerosolizes and increases downwind aerosol sea salt concentrations. Recent measurements, however, have suggested that frost flowers do not aerosolize (Yang et al., 2017). The frost flower hypothesis also does not hold for the results presented here because, while the $Cl^-:Na^+$ ratios are increased in the spring measurements, the ratios are not indicative of mirabilite sodium depletion. Other

work has suggested that blizzard and strong wind conditions (more common in winter than summer) may cause significant long range transport of sea salt aerosols (Hara et al., 2004). This hypothesis is more consistent with the 2ODIAC observations but it is still unclear as to what the exact mechanism would be, especially from continentally dominated air masses.

In addition to the seasonal differences observed for overall sea salt aerosol mass, seasonal differences in sea salt aerosol fractionation are also well documented. The springtime observations of enhanced Cl concentrations relative to Na, again while not at the level expected for mirabilite depletion, are consistent with previous observations (Jourdain and Legrand, 2002; Rankin and Wolff, 2003). However, it is unclear if the differences in the $Cl^-:Na^+$ ratios measured during 2ODIAC are due to $Cl^-$ or $Na^+$ gains or losses. The previous studies examining the Na/Cl fractionation have attributed these seasonal sodium depletions (or chloride enhancements) to frost flower formation (which promotes mirabilite formation) in the winter and early spring (Jourdain and Legrand, 2002; Rankin and Wolff, 2003; Hara et al., 2004; Kaleschke et al., 2004). However, as mentioned earlier, air masses in the spring passed predominantly over the continent (Section 3.2) and should therefore not have experienced a high degree of exposure to frost flowers. In contrast to the springtime chloride enhancement, the summer chloride depletion (sodium enhancement) has much better supported mechanisms to explain the sea salt fractionation in the aerosol phase. Multiple studies have suggested that $Cl^-$ liberation from aerosol to the gas phase is assisted by reactive nitrogen oxides (primarily $HNO_3$, $N_2O_5$, and $NO_3$; e.g. Finlayson-Pitts, 2003 and references therein), $H_2SO_4$ (Mouri et al., 1999), and organic acids (Laskin et al., 2012). In both field seasons, there are nitrates and sulfate in the filter measurements but neither appear to correlate with $Na^+$ or $Cl^-$ concentrations. Many of the proposed and measured reactions preferentially liberate $Br^-$ before $Cl^-$ which is consistent with the lack of observed $Br^-$ in either the filters or the AMS data (not shown). Additionally, since many of the proposed reactions are photochemical in nature, the elevated occurrence of $Cl^-$ liberation in Austral summer over spring would be expected (see SI Fig. S1). Unfortunately, measurements of gas-phase HCl were not available to confirm this mechanism. Beyond the differences in the seasonal overall and relative concentrations of $Na^+$ and $Cl^-$, no discernible patterns emerge when the data is analysed on other time scales. Analysis of the filters on intra-seasonal (day-by-day) timescales do not present any correlations, unlike aerosol-phase $SO_4$ (see Giordano et al., 2017).

In addition to the $Na^+$ and $Cl^-$ IC analysis of the filters, other relevant sea salt cations and anions were also analysed including $Br^-$, $NO_2^-$, $NO_3^-$, $SO_4^{2-}$, $Ca^+$, $K^+$, $Mg^+$, and $NH_4^+$. The results for these other ions fall broadly in line with the $Na^+$ and $Cl^-$ results with respect to overall concentrations; that is, for all measured ions the spring concentrations are higher than the summer concentrations. Just as with $Na^+$ and $Cl^-$, examining the overall ratios of the other measured cations and anions to each other can yield insight into potential processing differences between the seasons. One of the reasons that calculating non-sea-salt- (nss-) contributed ions is difficult in Antarctica is that almost none of the sea salt ions are truly conservative because of the ability to form fractionated salts at near- or below-freezing temperatures (e.g. ikaite, gypsum; see Hara et al., 2012 and references therein). This non-conservation yields aerosols that may be sea-salt derived but are not within sea-salt ionic ratios as discussed above with regards to mirabilite formation. Because of the importance of sea salt particles in heterogeneous reactions with reactive halogens, it is important to quantify what, if any, precipitation is occurring (Simpson et al., 2007). With regards to the $Mg^+/Na^+$, $Ca^+/Na^+$, and $K^+/Na^+$ ratios, no fractionation was evident in either field season. This is not in agreement

with results from Eom et al. (2016) which measured higher inorganic salt concentrations in summer aerosols by Raman microspectrometry and attenuated total reflection Fourier transform infrared imaging techniques. However, this result is slightly complicated by the fact that it is unclear if the $Na^+/Cl^-$ ratios are off due to $Cl^-$ depletion or $Na^+$ depletion but the fact that the cation ratios are all approximately equal to the ratios of sea water may suggest that the enhancements/depletions are

due primarily to $Cl^-$ and not $Na^+$. However, if there is not preferential $Cl^-$ depletion, then the rate of $Na^+$ depletion would have to equal the rates of depletion of the other cations in the aerosol phase. This seems unlikely but more work is needed to completely rule out the possibility.

The other anions were often below detection limits or not above the levels of blank filters. This is not unexpected due to the lower concentrations and multiple heterogenous and homogeneous reaction pathways available to aerosol phase ions

such as $Br^-$, $NO_x^-$, and $SO_4^{2-}$. One notable result, however, is the $SO_4^-$ enhancement, relative to $Cl^-$, measured in the summer filters. The springtime $Cl^-/SO_4^{2-}$ ratio is erratic, falling on both sides of the seawater ionic ratio. The summer filters, however, consistently exhibit $SO_4^{2-}$ enhancement. This agrees well with previous measurements and the well documented summertime sulfate increase over Antarctica that originates from biogenic activity in the Southern Ocean (see Giordano et al., 2017 and references therein).

**3.1.2 Snow Samples**

The snow samples (which consisted of blowing snow at ground level) show very different ion ratio patterns compared to the aerosol filters. Unfortunately, snow samples only exist from the spring 2015 field season so inter-seasonal conclusions cannot be drawn. Additionally, the time integrations of the snow samples are generally longer than those of the filters. In contrast with the springtime aerosol filters, which are either slightly $Cl^-$ enhanced or within 20% of the seawater ionic ratio,

the springtime snow samples are either slightly $Cl^-$ depleted or within 20% of the seawater ionic ratio (see SI Fig. S3). The total concentrations of ions in the snow samples, however, were two orders of magnitude higher than the total concentrations in the filters. The slight, but statistically significant, $Cl^-$ depletion in the snow samples is not evidence in and of itself that there is $Cl^-$ liberation from the snow to the aerosol phase. The loss of $Cl^-$ from the snow pack observed in early spring during 2ODIAC may be due to liberation to either the aerosol or gas phase as suggested by previous studies (e.g. Jourdain and Legrand,

2002). Unfortunately, the current data set is unable to prove or disprove such a hypothesis. The ratios of the other measurable ions, $Mg^+:Na^+$, $Br^-:Cl^-$, $SO_4^{2-}:Na^+$, all fall within ±20% of sea water. It should be noted that the conditions required for bromine activation (enhanced $Br^-$ and acidic pH for snow melt water; Pratt et al., 2013) were not present in any of the snow samples from 2ODIAC. The lack of bromide depletion, coupled with the $Cl^-$ depletion of the snow samples, is substantially different from what is typically observed in the Arctic springtime. It is also interesting to note that the snow and filter samples are

generally anti-correlated. When snow $SO_4^{2-}$ concentrations are at their highest, filter $SO_4^{2-}$ concentrations are at their lowest. Decoupling the snow and aerosol $SO_4^{2-}$ concentrations, however, is difficult due to the role MSA and aged, transported sulfate plays in the total sulfate budget for Antarctic aerosols (Giordano et al., 2017). Unfortunately, because snow samples were only available during one of the two field seasons, they should not be used to draw conclusions about the role snow plays in

the seasonal differences in the composition of Antarctic aerosols. However, the measured $Cl^-$ depletion does suggest that heterogeneous chemistry of HCl (and potentially $HNO_3$) may be occurring at the snow/atmosphere boundary.

## 3.2 Air Mass Back Trajectories and Wind Speed Analyses

Figure 3 shows the histogram and cumulative probability plots for wind speed and direction for the 2014 and 2015 field seasons. As briefly mentioned in Giordano et al. (2017), both field seasons differ in terms of both their wind direction and wind speed distributions, but both field seasons are consistent with historical records of winter/spring and spring/summer winds near Ross Island (Seefeldt et al., 2003). The winter/spring 2015 field season is dominated by winds coming out of the ESE (80% of the total record), usually at "high" wind speeds (60% of the record above $8ms^{-1}$). The spring/summer 2014 field season, by contrast, is dominated by lower wind speeds (80% of the total record below $8ms^{-1}$) with the direction distribution being 60%/40% out of the ESE/NW.

For the purposes of this manuscript, the local wind field at the 2ODIAC site is categorized into six wind regimes. The bimodal direction distribution creates two categories, which are further broken into "low", "medium", and "high" wind speeds using 0-1.99, 2-7.99, and 8+ $m\ s^{-1}$ bins, respectively. The threshold of the high wind speed bin was chosen because of both literature and observations supporting $8\ ms^{-1}$ as the point at which ground-based snow becomes lofted off the ground (Li and Pomeroy, 1997). The lower speed bins are broken into two categories to explore any differences that arise between calm or still wind conditions when strong stratification in the boundary layer would be expected, and light to moderate breeze conditions which would reduce this stratification. The effects of local wind conditions on Antarctic aerosol composition and total mass are discussed below.

The local wind conditions are necessary to discuss how Antarctic aerosols respond to effects such as blowing snow but back trajectory analysis on the air masses measured during 2ODIAC is necessary to provide context for potential aerosol sources. Back trajectory analysis was performed using NOAA's HYSPLIT tool using the GDAS 0.5º x 0.5º meteorology dataset at multiple altitude initializations (2m, 20m, 50m, 100m, 200m, 500m, and 1km AGL; Draxler and Rolph, 2015). Trajectories were initialized for 7 days at 3-hour time steps for the entire duration of both field seasons. The trajectories of the 2m AGL air masses fall into three general categories:

1) Continental: Air masses that travel solely over the continent to the measurement location
2) Marine: Air masses originating or predominantly traveling over the Southern Ocean (and sea ice)
3) Mixed: Air masses that travel predominantly over the continent to the measurement location but also travel over Southern Ocean open water and/or sea ice close to the measurement site

An example of each of the three categories is shown in SI Fig. S4. Three-hour initializations yield N=296 and 288 total back trajectories (37 and 36 days, respectively, at 3 hour intervals for each release height) for the Austral summer (2014) and spring (2015) field seasons, respectively. For the following analyses, these total trajectory numbers are assumed to be large enough to constitute a statistically random sampling of all of the air masses measured during 2ODIAC. It should be noted that the 0.5º x 0.5º grid in the dataset does not adequately capture the effects of local orography on air masses. Additionally, the 2m AGL

altitude initialization for HYSPLIT may not be highly accurate in this dataset. However, there are few differences in the 2m and 50m initializations aside from vertical trajectories the air parcels follow so the 2m initializations are used to reflect the station sampling height. The categories presented here should therefore be read as a generalized representation of the air mass paths, as many of the air masses would have had some degree of travel over sea ice (therefore including some open-water polynyas) to the measurement location. Over the course of both field seasons, the majority of the initialized back trajectories are classified as predominantly continental (Category 1; 60% in Summer 2014, 80% in Spring 2015). Predominantly marine and mixed air masses (Category 2 and 3, respectively; both 20% in 2014, 10% in 2015) were much less represented in the data. Interestingly, despite the inability of the 0.5º grid to capture local orography effects on air masses, the HYSPLIT results fall broadly in line with the measured wind directions at the field site.

The HYSPLIT results, as a whole and the examples in Fig. S4, indicate a substantial amount of vertical mixing is commonplace over all of the air mass altitudes initialized in the back trajectories. Descent of air masses from over 4000m AGL to 50m or 100m AGL over the trajectory time periods was not uncommon for air masses that travel over the continent itself. There is also a general trend of substantial vertical movement of air masses between 0-1000m AGL for these continental (category 1) air masses. The mixed air masses (category 3), are generally similar in terms of vertical movement. For air masses that originate or travel predominantly over the ocean/sea ice (category 2), there is much less vertical mixing with most of the sub-100m air masses staying at or below 100m AGL for the entirety of their trajectories.

### 3.3 Aerosol Number and Size Distribution Changes due to Air Mass and Wind Speed Regimes

Two major conclusions of this manuscript are that blowing snow plays a major role in both aerosol number and composition in Antarctica. However, blowing snow must be measured for this conclusion to be well supported. To measure blowing snow during 2ODIAC, a laser disdrometer which measures the number, size and vertical velocity of hydrometeors from 0.2 – 25mm diameter was used. For these measurements, the hydrometeors were entirely snow particles which were primarily moving horizontally, and thus the sizing and vertical velocity measurements are uncertain. However, the instrument accurately reports the number of particles that pass through the sample volume in a given time and thus the particle counts are the convolution of the snow particle density as well as the velocity of the particles, which in turn is a function of wind speed. Figure 4 shows the laser disdrometer snow particle number counts as a function of wind speed. The curve of the data suggests that there are at least three regimes in which snow particles exist over the measurement site: one regime below approximately 5.5 ms$^{-1}$, a transitionary regime between 6-12 ms$^{-1}$, and one regime above 12 ms$^{-1}$. With wind speeds below 5.5 ms$^{-1}$, blowing snow counts rarely exceed 5 snow particles per 10 seconds. Therefore, at low to medium wind speeds, snow was rarely lofted high enough to enter the disdrometer optical path (approximately 5m above surface level). Between 6.5 to 12 ms$^{-1}$ a significant non-linear increase in the number of measured snow particles exists. Above 12 ms$^{-1}$, the number of snow particles increases in a steeper, but largely linear, relationship to wind speed. The low wind speed snow particles are primarily non-wind generated snow particles such as falling snow and diamond dust. Near the transition wind speed, blowing snow lofted from the surface

begins to reach the height of the instrument and the increase is non-linear as particle counts are increasing both from increasing wind and increasing particle number density as the blowing snow layer envelops the instrument. Finally, at the highest wind speeds, the disdrometer is entirely within the blowing snow layer and the increase in particle counts is primarily due to increasing winds speed and particle velocity. This is in good agreement with previous observations and of estimates of the

wind speed threshold for blowing snow (Li and Pomeroy, 1997). This also lends strength to the "high" wind speed bin classification (> 8 m/s) used in the analysis presented here.

Recent work in the winter in the Weddell Sea has suggested that blowing snow plays a major role in aerosol number density in the high latitudes (e.g. Yang et al., 2016). The role of blowing snow in aerosol generation is further supported by the observations presented here during 2ODIAC. Figure 5 shows particle counts in different size ranges from the Lighthouse

OPC as a function of wind speed. Below 8 ms$^{-1}$, aerosol number concentrations are flat and consistent in all measured size bins. Above 8 ms$^{-1}$, particle counts in all size bins increase nearly linearly with increasing wind speeds. Aerosol number concentrations respond to changes in wind speed within minutes, before, during, and after high wind events (SI Fig. S5). If the aerosol number distributions are analysed as a function of wind direction or air mass back trajectories, no significant differences arise. These results are somewhat inconsistent with what Hara et al. (2014) observed over Swoya station in late Austral winter.

The results presented by Hara et al. (2014) showed a period of enhanced aerosol number concentrations (primarily in the 500nm - 2μm size ranges) arising during a low wind speed period and, overall, that significant air mass travel over sea-ice increased aerosol number concentrations. It is unclear if the period of decreased wind with enhanced aerosol concentrations was a local meteorological phenomenon that was simply not observed during 2ODIAC. Likewise, it is unclear if the travel over ocean/sea-ice observed in 2014 by Hara et al. generated aerosols via the same mechanisms observed during 2ODIAC

because compositional information does not exist during those measurements.

Some work has suggested that fragmentation of wind-blown snow crystals may be significant in generating smaller particles from large snow crystals (Comola et al., 2017), and it is tempting to explain the increases in particles from blowing snow in this manner. Tests performed in the field, however, suggest that snow-fragmentation in the inlet is not what was measured during 2ODIAC. A series of tests were conducted during moderate and high wind speeds where the inlet heater was

both turned off and scanned through a range of temperatures. Results from these tests showed that counts were generally consistent across heater temperatures. The increases in particle counts were therefore neither due to evaporating snow (and thus counting internally mixed condensation nuclei) nor due to fragmented snow particles. Unfortunately, the measurements made cannot identify the exact mechanism of particle generation by blowing snow; more detailed study of the microphysics is required. Additional measurements and targeted study of the process is required.

Figure 5 (and SI Fig. S5) show that blowing snow has major implications for aerosol transport and lifecycles. Because observed aerosols rapidly decrease with decreasing wind speeds at the ends of a blowing snow event, it is likely that larger (supermicron) aerosols are rapidly redeposited to snowpack. This deposition and subsequent re-lofting of aerosol may explain longer range transport of chemical species (found primarily in the aerosol phase) to inland continental snow. Additionally, this clearly represent a pathway from ice/ocean of aerosols to the atmosphere. While these measurements clearly show that blowing

snow is a dominant driver of aerosol production in the boundary layer, it is unclear what portion of these aerosol are transported into the free troposphere and what their larger scale impact may be.

### 3.4 Chemical Composition of Antarctic Aerosols by Air Mass Trajectories

One of the unique traits of the 2ODIAC deployment of an AMS is the high temporal resolution of the instrument (approx. 2-minute data records). This high temporal resolution allows for composition data of the submicron aerosol to be analysed in the context of rapidly changing meteorology and other short term phenomena. While there has been work analysing aerosol filters in terms of back trajectory analysis (e.g. Hara et al., 2004), the long time integrations (24 hours) of the filters can hide the effects that rapidly shifting air masses and local meteorology may have on the aerosol population, and are mass weighted for events that have higher aerosol loadings (e.g. higher wind speeds). The results presented in this section use the air mass back trajectory analysis of Section 3.2 to parse the AMS data in a way that better informs the effects of air mass back trajectories on aerosol composition.

Figure 6 shows, for both field seasons, the mass of the 5 main AMS speciation components - chloride, sulfate, organics, nitrate, and ammonium - along with the estimated Na mass as functions of the three main back trajectory categories defined in Section 3.2. Both $Na_{AMS}$ and the chloride (here referred to as "Chl" to be consistent with AMS terminology) mass concentrations have been calculated based on the process described in Section 2.3. The data in Fig. 6 is parsed into both field seasons because, (with seasonal differences in mind) the back trajectories of the air masses can be compared inter-annually as well as intra-annually. From Fig. 6 it is clear that there are no statistically significant differences in Chl:$Na_{AMS}$ intra-seasonally, i.e. between different air masses for a given measurement campaign season (average Chl:$Na_{AMS}$ = 1.9 ± 0.3 for the 2014 spring/summer and 1.0 ± 0.1 for the 2015 winter/spring measurements; all values shown in Table S1). It is unclear why the continental air masses measured in the 2015 winter/spring season are higher than the marine and mixed air masses but the Chl:$Na_{AMS}$ ratio is consistent within measurement and propagated error. The absolute concentrations of both ammonium and nitrate also remain consistent within measurement error across the various air mass types for a given campaign season. This is inconsistent with previous measurements over similar size distributions by impactors by Virkkula et al. (2006). Organics concentrations do show small variations in the summer marine air masses (10.4 ± 1.4 ng m$^{-3}$) as compared to the other summer air masses (2.6 ± 0.8 and 2.1 ± 0.5 ng m$^{-3}$ continental and mixed, respectively) which is unsurprising considering summer oceanic phytoplankton activity. Conversely, lower organics concentrations in the marine spring air masses were also observed (0.2 ± 2.5 as compared to 10.1 ± 2.0 and 4.7 ± 2.2 ng m$^{-3}$ for the continental and mixed air masses, respectively). It is unclear why organics are so much higher in the continental spring air masses but may be related to sustained high winds over the continent lofting and transporting previously deposited organics from the previous seasons. The higher organics may also simply be due to higher energy requirements for the human settlements on the continent in the winter. More work is needed, however, to determine if this is a sustained trend or not. Sulfate concentrations are also higher in the continental air masses for both seasons as compared to the marine and mixed air masses and are likely related to the results discussed in Giordano et al. (2017).

The lack of intra-seasonal variation in the data suggests two possibilities: that air mass back trajectories have a small impact on aerosol composition, at least over coastal Antarctica, and/or that even short transit periods over sea ice (every air mass measured during 2ODIAC had to transit over at least 8km of sea ice) can have strong impacts on aerosol composition. The latter possibility may help explain the divergence of the 2ODIAC data from previous measurements - such as the

observation that ammonium was observed to be higher in the summer than spring at other coastal Antarctic sites (Legrand et al., 1998). An increased number - or closer proximity - of penguins and seals to the field site would potentially contribute to higher ammonium and organic source emissions and, by extension, aerosol concentrations. Given the closer ice edge of the spring field season and anecdotal observations of more penguin activity near the field site in spring, this is a possible explanation for why the 2ODIAC $NH_4$ and organics concentrations differ between the 2014 and 2015 datasets (Schmale et al.,

2013). A similar argument can be made for nitrate, which showed higher spring concentrations than summer concentrations. Previous measurements have observed the opposite trend in nitrate (Wagenbach et al., 1998b). The main sources of nitrate to Antarctic aerosols are $NO_x$ cycling in snowpack and nitrate sedimentation from polar stratospheric clouds (PSCs), the former of which is more important in October-December, the latter in July-September (Savarino et al., 2007). Since the snowpack driven nitrate would be most important during 2ODIAC and since most of the ice was at least partially snow covered, it may

be that the nitrate concentrations support the hypothesis that short transit over sea ice can strongly impact aerosol composition. A similar argument can be drawn for the organics measured by the AMS as well, assumed to come from Southern Ocean biota or even by biota found on the continent itself (Friedmann, 1982; Cochran et al., 2016). Unfortunately, testing such a hypothesis would require a spatially distributed measurement network which 2ODIAC was not. However, it is important to note what the emission sources are for a given aerosol species and how they do or do not fit in the air mass back trajectory analysis. The

former possibility, that air mass paths are less important than wind speed conditions, is part of the motivation for Section 3.5 and is examined there in depth.

Other aspects of the figure - e.g. higher $Na_{AMS}$ and Chl concentrations in the spring, higher sulfate concentrations in the summer - are well supported by the filters (Section 3.1) and/or by previous work (e.g. Rankin and Wolff, 2003). One unsupported result from the figure, however, is the statistically significantly higher sulfate concentrations in summer

continental air masses. The prevailing hypotheses suggest that most of the increased sulfate aerosol in Antarctica in the spring/summer comes from DMS oxidation products (e.g. Beresheim and Eisele, 1998; Jourdain and Legrand, 1999). One may therefore expect the marine, or even mixed, air masses to have higher sulfate concentrations than the continental air masses. A possible explanation for the higher continental sulfate concentrations could be an outflow of previously snow-deposited sulfate aerosol from inland earlier in the year (Harder et al., 2000).  As wind direction is highly correlated with wind speed,

with higher wind speeds associated with air masses approaching from the SE (continental interior), it is also difficult to separate the effects of air mass trajectory from wind speed (Section 3.5). However, this is still consistent with air mass origin playing a less significant role in aerosol composition than other factors. The higher concentrations could also simply be an artifact of the fact that the summer (2014) field season had more continentally-originating air masses and the marine and mixed air mass trajectories are simply under-sampled.

The most noteworthy result of the back trajectory analysis is that there are few significant differences in absolute or relative concentrations of the Antarctic submicron aerosol mass, either on an inter- or intra-seasonal basis, when viewed as a function of air mass history. These results are in contrast to recent results from Concordia, Antarctica which showed that air masses traveling over sea ice, as opposed to open-ocean, had significant sulfate depletion relative to sodium (Legrand et al., 2017). However, drawing a direct comparison between the Concordia results and these results is difficult for a number of reasons which are discussed in the next section. If air mass back trajectories do not explain composition differences from this study, then the question becomes what meteorological or other factors have the strongest influence on Antarctic aerosol composition. The following section indicates that wind speed plays a significant role in modulating the concentration and composition of Antarctic aerosols.

## 3.5 Chemical Composition of Antarctic Aerosols by Wind Speed Regime

Air mass back trajectories show that submicron aerosol composition does not appear to differ by region of origin or that differences are small and are masked by the meteorologically driven compositional changes. This result suggests that aerosol composition - over coastal Antarctica at least - may be controlled by processes that are independent of air parcel trajectory. During the 2ODIAC field seasons, aerosol number concentrations are closely tied to wind speed (Section 3.2). This result has prompted further investigation into how wind speeds affect the chemical composition of Antarctic aerosols. Here, we examine how the amounts and relative ratios of the major AMS-measured aerosol constituents, and the form of those constituents, are strongly tied to wind speed.

Figure 7 (and SI Fig. S6) shows the major aerosol components from the AMS as a function of the six wind regimes (not back trajectory classifications) defined in Section 3.2 for both field seasons. As presented in Fig. 7, the total estimated Chl and $Na_{AMS}$ are calculated as described in Section 2.3. Three major observations are apparent in Fig. 7 (and SI Fig. S6): first, that within a season, the absolute amount of chloride increases at high wind speeds only, with little variation between low and medium wind speeds; second, that the absolute amount of organics and sulfate increase with wind speed in the springtime measurements only; and third, that the absolute amounts of nitrate and ammonium do not statistically significantly respond to wind speed and are relatively independent of wind direction. For the 2014 spring/summer measurements, nitrate ranges from 1.6-2.6 ng m$^{-3}$ with a standard deviation of $\pm 1$ ng m$^{-3}$ and ammonium ranges from 5.0-9.0 ng m$^{-3}$ with a standard deviation of $\pm 2.0$ ng m$^{-3}$. For the 2015 winter/spring measurements, the values are 3.8-5.6 $\pm$ 2 ng m$^{-3}$ and 17-21 $\pm$ 3 ng m$^{-3}$ for nitrate and ammonium, respectively.

The increase of chloride as measured by the AMS with high wind speeds, especially without any air mass back trajectory dependence as discussed in Section 3.4, shows that submicron aerosol chloride concentrations are a strong function of wind speed. This result helps explain much of the variance in Fig. 6 since each air mass is an integration of all wind speeds. It also provides a potential explanation for why the supermicron filter measurements had higher Cl$^-$ concentrations in spring relative to summer (see Fig. 2). What this result does not do, however, is explain either the source of the aerosol chloride or

the mechanism by which it enters the aerosol phase. One explanation could be that high wind speeds loft snow, which also lofts and liberates any aerosols that may have deposited onto the snow pack. However, the lack of size distribution changes (e.g. Giordano et al., 2017) during high wind events would suggest that the aerosol size distribution is an intrinsic property in the snow pack, which seems unlikely but cannot be ruled out by this data set. Another possible explanation for the increased

chloride concentrations is that lofted snow is mechanically fractured (possibly through snow-snow collisions) or sublimated into submicron aerosols. This explanation, however, would also cause a disproportionate rise in sodium in the submicron aerosol population at least in the springtime measurements since there is $Na^+$ enhancement in the snow (see Section 3.1.2). Since no such disproportionate $Na_{AMS}$ increase is noted by the AMS, a mechanical fracturing or sublimation of snow into smaller aerosols also seems unlikely. The final possible explanation is that blowing snow creates or catalyses a chemical

mechanism which increases aerosol chloride from the supermicron aerosol population that then partitions to the submicron.

The possibility of a chemical mechanism to increase aerosol chloride occurring during blowing snow events is supported by existence of fast vaporizing chloride (Section 2.3). The increase in relative fraction of total chloride associated with fast vaporizing chloride and $W_xO_yCl_z$ complexes during high wind events is shown in Fig. 8. Figure 8 shows the percentage of the total reconstructed chloride signal by the AMS that is from the fast vaporizing chloride as a function of wind

speed. The immediate increase at 8+ $ms^{-1}$ coincides with the "high wind" speeds in Fig. 7. The fact that not only is the overall aerosol chloride increasing during blowing snow events but that the chemical form of that chloride is changing is strong evidence for a physical or chemical mechanism altering aerosol composition during blowing snow events. The fact that there is an increase in fast vaporizing chloride suggests that the increased chloride concentrations as measured by the AMS are not solely due to increase sea-spray aerosol transport. If sea-spray aerosol were the sole cause, then the ratio of fast:slow vaporizing

chloride would be expected to stay relatively constant since sea-spray aerosol would have also been measured at low wind speeds due to the location of the field sites relative to the ice edges. It is unclear from the data of 2ODIAC what the exact mechanism of chloride alteration may be. One hypothesis is that blowing snow (which also generally increased ambient relative humidity) adds aerosol which may have some associated liquid water which then disassociates some NaCl out of its crystalline form (akin to the "disordered interface", Bartels-Rausch et al., 2014). Vaporization of this liquid aerosol may contribute to

the "fast-vaporization" signal. Laboratory testing of wet vs dry NaCl aerosol support this as a possible mechanism. Additionally, the fact that Chl shows a depletion compared to $Na_{AMS}$ at high wind speed suggests the possibility of chemical reactions that liberate chloride from the submicron aerosol. Chloride in the submicron aerosol may therefore be in more labile forms in the aerosol phase such as $NH_4Cl$ or as HCl. These species are non-refractory and would flash vaporize in the AMS and lead to increases in the fraction of fast-vaporizing chloride. It may also be possible that reactions at the aerosol surface

produce volatile Cl-containing gases which then deposit on the submicron aerosol. This pathway could also produce results consistent with the measurements. Regardless of the actual pathway of Cl mobility, it is important to note that the majority of the chloride signal still remains as a semi-refractory component. However, the clear increase in the percentage of fast vaporizing chloride fraction is apparent with higher wind speed, and most pronounced in the 2014 dataset.

In contrast to the chloride concentrations and their wind speed/blowing snow dependence, Fig. 7 (and Fig. S6) clearly shows that nitrate and ammonium show no such dependence. In the case of nitrate, this is especially interesting given the importance of snowpack $NO_x$ on aerosol nitrate concentrations during the 2ODIAC sampling period (Savarino et al., 2007). It might be expected that high wind conditions would create a concentration gradient over the snow pack encouraging emission

and, supposedly, condensation onto existing aerosols. However, the $NO_x$ snowpack cycling is dominated by either a photochemical process or a thermodynamic equilibrium process (Röthlisberger et al., 2002). Since high wind speeds were generally accompanied by cloud cover, the photochemical process can be assumed to be negligible for this study. The thermodynamic process has also been shown to have a negative temperature relationship with peak emission rates occurring between -60°C to -30°C, far colder than experienced even in the springtime measurements of 2ODIAC. Taken together, the

lack of wind speed dependence for nitrate should therefore not be surprising. Ammonium is slightly different though with these results suggesting that ammonia processing in this area is not modulated by wind speed.

It should be noted that $NO_x$ snowpack cycling was observed in the summer 2014 field season under low wind (and generally cloud-free) conditions (see SI Fig. S7). What is interesting is that, despite the clear gas-phase $NO_x$ cycling, corresponding increases or cycles in aerosol nitrate concentrations were not observed. Legrand et al. (2017) suggests that

nitrate/nitric acid plays a significant role in acidifying sea-salt particles over the Antarctic plateau, which displaces aerosol chloride. Such a correlation is not observed in this data set. Results from Legrand et al. (2017), however, indicate that the nitrate measured over Dome C, Antarctica may be due to transport via dust aerosol and not due to snowpack $NO_x$ cycling. The lack of crustal ion enhancements in the filter samples suggests that mineral dust does not play a large role over the McMurdo area sea ice and therefore the nitrate source assumed for Dome C is not available for chloride depletions around the McMurdo-

area sea ice.

### 3.6 The Antarctic Halogen Cycle: AMS Derived Chloride to Sodium Ratios as a Function of Wind Speed Regime

The previous section demonstrated that, for a given season, overall mass of chloride and sodium are strongly dependent on local meteorological conditions. Harder to parse from Fig. 7 is if high wind speeds disproportionately affect either chloride or sodium concentrations in submicron aerosols. Recent results from Legrand et al. (2017) show that substantial chloride

depletions occur over Dome C, Antarctica with depletion peaking in spring, assumedly due to photochemistry and the recovery of acidic species in that Antarctic atmosphere. Figure 9 shows the average chloride to sodium ratios of both field seasons of 2ODIAC as a function of wind speed. One of the most interesting results from Fig. 9 is the apparent unimportance of mirabilite with Chl:Na$_{AMS}$ ratios as measured by the AMS never going over that of seawater. Even if the analysis is restricted to examining continental winds, the average chloride to sodium weight ratios in submicron aerosols never reach mirabilite depletion ratios

(see SI Fig. S8). These results, along with the bulk results from the filters, snow, and bulk submicron data from the AMS (Sections 3.1-3.5), suggest that mirabilite formation along coastal Antarctica may only have episodic impacts on the Antarctic troposphere and not a regional significance.

Figure 9 also shows that, generally, submicron aerosols in coastal Antarctica are subject to similar degrees of chloride depletion as has been previously reported (Wagenbach et al., 1998; Hara et al., 2002; Jourdain and Legrand, 2002; Legrand et al., 2016; Legrand et al., 2017) with the important distinction that these measurements are online, high time resolution as compared to previous studies which are all filter based. Wind speeds below 7 ms$^{-1}$ show the least amount of chloride depletion

with Chl:Na$_{AMS}$ molar ratios >0.8 in summer and >0.5 in spring. Above 7 ms$^{-1}$, Chl:Na$_{AMS}$ ratios fall in both seasons to ~0.7 in summer and ~0.5 in spring. At extremely high wind speeds (>13 ms$^{-1}$), the molar ratio reverses direction and begins to increase. This is near the same wind speed at which snow particle numbers increase more rapidly (7-13ms$^{-1}$; Fig. 4). Surface-layer snow is generally the same Cl$^-$:Na$^+$ molar ratio as seawater at relatively high concentrations (SI Fig. S3), which may be contributing to this change in the submicron Chl:Na$_{AMS}$ molar ratios but the cause remains unclear. Again, this relationship of

Chl:Na$_{AMS}$ to wind speed holds generally true for examinations of only winds from the continent or sea ice (SI Fig. S8). Fig. 9 supports the rest of the results presented here that suggests that high wind speeds (>7ms$^{-1}$) subject air parcels to conditions which modify either the mechanisms or rate of chemical processing for Antarctic aerosols compared to lower wind speeds.

The relationship between Chl:Na$_{AMS}$ in the submicron aerosol population and wind speed further suggests that blowing snow itself is indeed a significant source of both sodium and chloride to the Antarctic troposphere. The remaining

question is simply whether blowing snow physically or chemically alters the chlorine found in the snowpack and to what extent. Combining Figs. 7, 8, and 9 demonstrates that blowing snow strikingly increases the overall amount of both Na and Cl to the submicron aerosol mass, alters the Cl phase inside the submicron aerosol, and is potentially driving Cl depletion. This work adds to the work of Lieb-Lappen and Obbard (2015) which showed that, in blowing snow only, Br$^-$/Cl$^-$ratios are altered with respect to surface layer snow. Further work is necessary to determine the exact mechanism through which blowing snow

drives composition changes but these results suggest that the total halogen budget of (coastal) Antarctica is dominated by local meteorological conditions that must be accounted for.

### 3.7 The Antarctic Halogen Cycle: Aerosol Iodine, Bromine, and Ozone

In addition to chloride, understanding the gas and aerosol phase behaviour of other halogens is important due to halogens' large impacts on atmospheric processing pathways (e.g. Simpson et al., 2007). Unfortunately, the aerosol phase

bromine and iodine pictures are not as clear as the aerosol chloride. Both species were generally at or above the detection limit of the AMS of 0.04 ng m$^{-3}$. There is some indication that Br is slow vaporizing in the AMS (see SI Fig. S9) but the slow vaporizing correction is not performed on Br here because the decay constants are not able to be determined from the available data. Therefore no quantification of the amount of aerosol-phase Br (nor I) is offered here, but rather this analysis is presented as evidence that the halogens do exist in an aerosol form, and are measurable with the AMS. Neither Br nor I show any

statistically significant dependence with air mass back trajectory, wind direction, or wind speed. However, over both field seasons and for both halogens, aerosol phase concentrations are at a local maxima at wind speeds 0-2 ms$^{-1}$, decline to a local minima from 3-6 ms$^{-1}$, and rise to another local maxima at 7+ ms$^{-1}$. However, the overall concentrations of the two halogens are inversely related over the two field seasons: aerosol bromine is higher than iodine in the spring/summer (2014) while iodine

is higher than bromine in the winter/spring (2015). This is not consistent with gas-phase measurements of IO and BrO made by Saiz-Lopez et al. (2007) at another coastal site in Antarctica. These results suggest that the mechanisms of transport into and out of the aerosol phase may have a significant seasonal aspect, which is consistent at least with the literature regarding gas phase halogen species in the Arctic and Antarctic (Simpson et al., 2007, 2015). In the Arctic, research has shown that this

seasonality in bromine and iodine is unlikely to be controlled by temperature (Halfacre et al., 2014), though may play a part in chlorine production in snowpack (Custard et al., 2017). Furthermore, photolytic reactions are well known to produce reactive bromine chemistry in the Arctic (Burd et al., 2017). For the Antarctic, it is likely that the same mechanisms control the halogen cycle but further research is needed.

As a bulk measurement, mass fractions in the filter measurements are strongly biased towards larger, supermicron

particles, whereas the AMS measurements are only sensitive to submicron particles. The below LOD concentrations of $Br^-$ in the filters combined with measurable Br with the AMS is generally consistent with measurements of the cycling of inorganic bromine in the marine boundary layer showing depletions of bromine in supermicron aerosol and enrichments in submicron aerosol (Sander et al., 2003). These results are also consistent with measurements at another coastal site in Antarctica (Legrand et al., 2016). This apparent size dependence for bromine in the aerosol phase supports models such as Legrand et al.'s (2016)

that implement size-dependent depletion factors for bromine from aerosols. Additionally, the AMS measured Br as a function of wind speed results support the hypothesis that blowing snow provides a source of bromine. The local maxima of submicron Br concentrations above the wind speed threshold for blowing snow may suggest that gas-phase bromine is being liberated and absorbing or heterogeneously reacting onto the submicron aerosol population. Unfortunately, the lack of concurrent gas-phase bromine measurements means these results are not conclusive evidence for these hypotheses but do point out the need

for more measurements.

The results presented here suggest that source regions for air masses do not significantly affect aerosol halogen concentrations in Antarctica. However, we do not here assess what, if any, impact that vertical displacement during transport may have on the halogen concentrations in the aerosol phase. Due to the low concentrations of aerosol-phase halogens measured, the uncertainty in the HYSPLIT runs, and the fact that the air masses examined in most of the HYSPLIT runs had

large altitude changes during their back trajectory lifetimes, assessing the effect of vertical displacement of an air mass on halogen or aerosol concentrations or compositions is outside of the scope of this manuscript. However, it should be noted that models have shown there to be significant differences in the vertical distribution of gas-phase halogens in the boundary layer (Lehrer et al., 2004; Saiz-Lopez et al., 2008). Any vertical movement of an air mass could therefore have significant impacts on the halogen cycle and halogen-aerosol interactions.

Another significant aspect of the aerosol halogen content is that it is sometimes correlated and sometimes anti-correlated with gas-phase ozone concentrations. Figure 10 shows an example of both scenarios from the early summer season of 2014. On November 15, 2014, aerosol iodine increases significantly as gas-phase ozone decreases from 26 ppb to 17 ppb while aerosol bromine shows increase over background for the same period. On November 19, 2014, both iodine and bromine increase significantly concurrent with an increase of ozone from approximately 22 ppb to 27 ppb. Both iodine and bromine

also follow a spike in ozone concentrations after a slight decrease from the initial ozone burst. Both events occurred during low to moderate (0-5 ms$^{-1}$) wind speeds predominately from wind out of the east/southeast. It is therefore unclear what relation the submicron aerosol halogen content has to ozone, in particular ozone depletion events. Previous work, especially in the Arctic, has well established anti-correlations between filterable Br$^-$ and ozone as well as correlations between Br$^-$ and I$^-$ (e.g.
Barrie et al., 1994) but not in the temporal resolution that is available in this study. It therefore remains difficult to contextualize these results, and, as with the chloride observations, gas phase measurements of halogens and halogen oxides would be necessary to propose a mechanism to reconcile these two disparate observations.

## 4 Conclusions

The results presented here suggest that the chlorine and sodium fraction of Antarctic boundary layer aerosol is strongly
affected by local meteorological conditions. The results presented here further suggest that air mass origin has a much smaller impact on aerosol composition and concentration relative to the impact of local meteorology. While the results here are limited to measurements on snow covered sea ice, it should be noted that during the spring and early summer the area of sea ice is typically larger than the total land area of Antarctica (Comiso et al., 2016), and thus the aerosol production related to blowing snow may have a major regional impact. Additionally, ionic concentrations in blowing snow samples and super-micron aerosol
mass suggest that the submicron halogen cycle is a process that is distinct from the supermicron and snow-phase halogen and sodium concentrations. The 2ODIAC measurements provide evidence that there are one or more aerosol sources or processing pathways that have strong impacts on aerosol mass.

Despite the novel measurements presented here and the coincident submicron aerosol, supermicron aerosol, and snow composition measurements, the nature of the halogen budget and cycle over Antarctica is still unclear. Recapping Sections
20    3.1-3.7:

1)  Snow samples (from Spring 2015) show anywhere from non-existent, to minor, to significant Cl$^-$ depletion, relative to Na$^+$.

2)  Aerosol filters collected in Spring 2015 show higher absolute amounts of both Na$^+$ and Cl$^-$ compared to summer 2014. This strongly agrees with the submicron AMS data for both seasons.

3)  Spring filters generally range from no Cl$^-$ depletion to slightly Cl$^-$ enhanced. Summer filters show a strong Cl$^-$ depletion.

4)  Blowing snow begins to occur between 6-10 ms$^{-1}$ and substantially increases aerosol number concentrations across the entire size distribution. Aerosol number concentrations are strong functions of wind speeds.

5)  Submicron AMS measurements show *more* Chl depletion in the spring (average Chl:Na$_{AMS}$ = 0.8) than in the summer
(average Chl:Na$_{AMS}$ = 1.25) but both seasons are generally depleted in Chl relative to Na$_{AMS}$ as compared to seawater ratios.

6) Based on AMS measurements, low wind speeds ($<3ms^{-1}$) are much more likely to show *no depletion* in Chl relative to seawater. High wind speeds ($>7ms^{-1}$) are almost *always statistically significantly depleted* in Chl, with an exception of a possible frost flower (or other unknown) event captured in Spring 2015.

7) Aerosol iodine and bromine content in the submicron aerosol show opposing trends between the spring and summer seasons. In the supermicron filters, both iodine and bromine are higher in the spring than in the summer.

8) Submicron aerosol iodine content can be either correlated or anti-correlated with ambient ozone concentrations and both iodine and bromine do not have any *statistically* discernible wind speed dependence, though a weak dependence may be evident.

These eight observations do not paint a clear picture of how halogens, specifically chlorine (but analogies may extend to bromine and iodine), cycle in Antarctica. These results make clear the need for additional measurements including gas-phase halogen species and single-particle measurements. A clearer understanding of the halogen cycle over Antarctica will result in better modelling capabilities of the oxidative capacity of the polar troposphere which impacts both current climate (e.g. DMS oxidation) and paleoclimate reconstructive abilities. The results presented suggest that the underlying mechanisms controlling chloride concentrations affect the sub- and super-micron aerosol populations differently. This conclusion is not wholly inconsistent with the results presented in Legrand et al. (2017) though the actual trends in Cl:Na ratios do not agree between these two studies. It is important to note that Legrand et al. (2017) is measuring at an inland site while these measurements took place at a coastal site.

There are a number of potential factors that influence the aerosol chloride concentrations. Wind speed (demonstrated here), photochemical reactions, snowpack chemistry, air mass hysteresis (demonstrated to not be a factor here but is in Legrand et al., 2017), tropospheric mixing, physical ice phenomena such as brine layers or frost flowers, and any combination of these are only a few potential candidates that may control the chlorine budget over length scales ranging from coastal Antarctica, to Antarctica as a whole, to the cryosphere as a whole.

**5 Data Availability**

Data are available at https://osf.io/sbw5j/

**Acknowledgements**

The authors of this paper would like to thank the National Science Foundation for funding this work through grant numbers 1341628 and 1341492. Additionally, the authors extend many thanks to all of the support staff at McMurdo Station, especially Tony Buchanan; without their support none of this work would have been possible. The authors would also like to thank Terry Deshler, Andrew Slater, and Anondo Mukherjee for their direct help with measurements in the field and Erin Frolli for her

assistance in satellite image retrieval and creation. Any opinions, findings, and conclusions expressed in this material are those of the authors and do not necessarily reflect the views of the NSF.

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

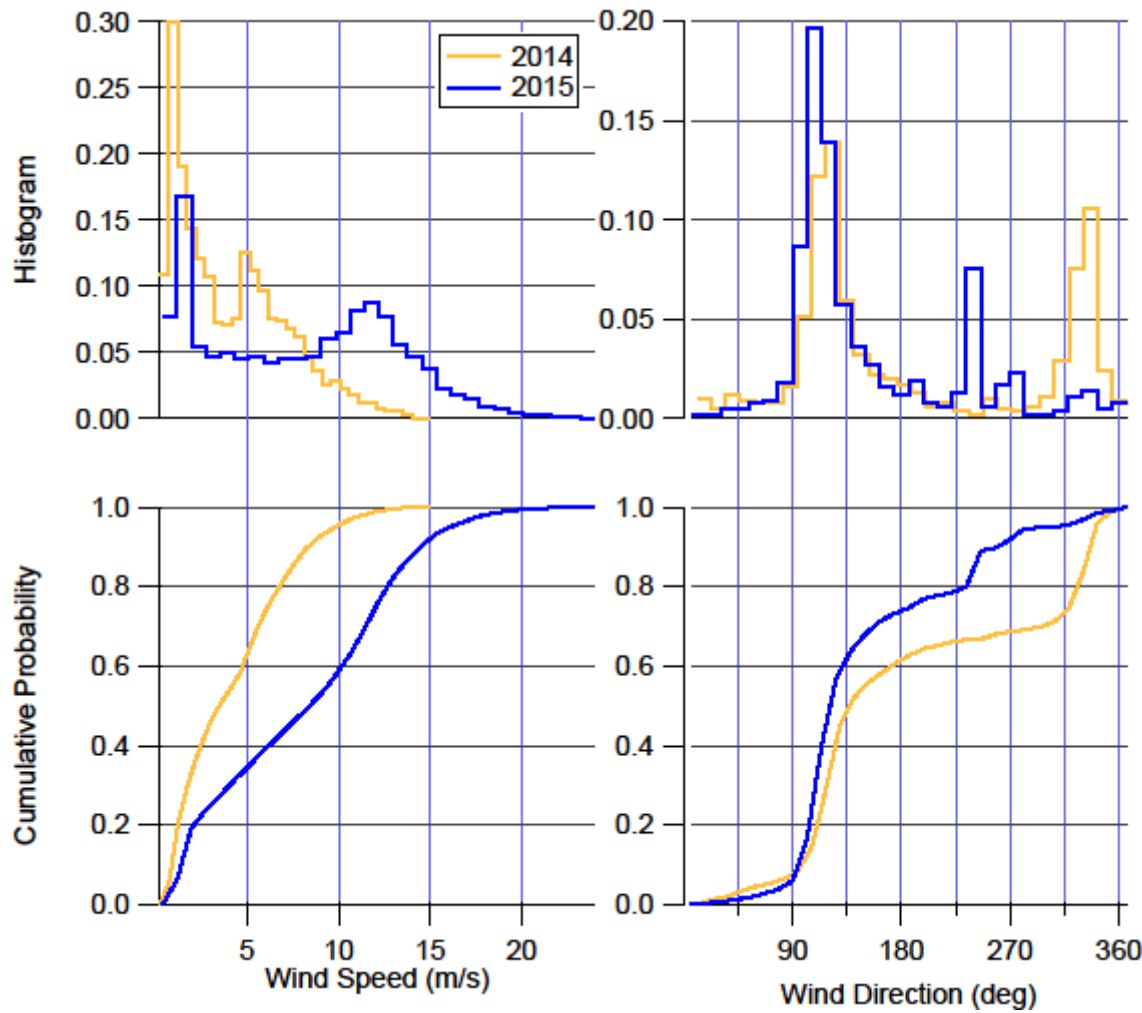

**Fig3 – Histogram and cumulative probability distributions for wind speed (ms⁻¹; left) and wind direction (deg; right) for both spring/summer field season (gold) and winter/spring field season(blue).**

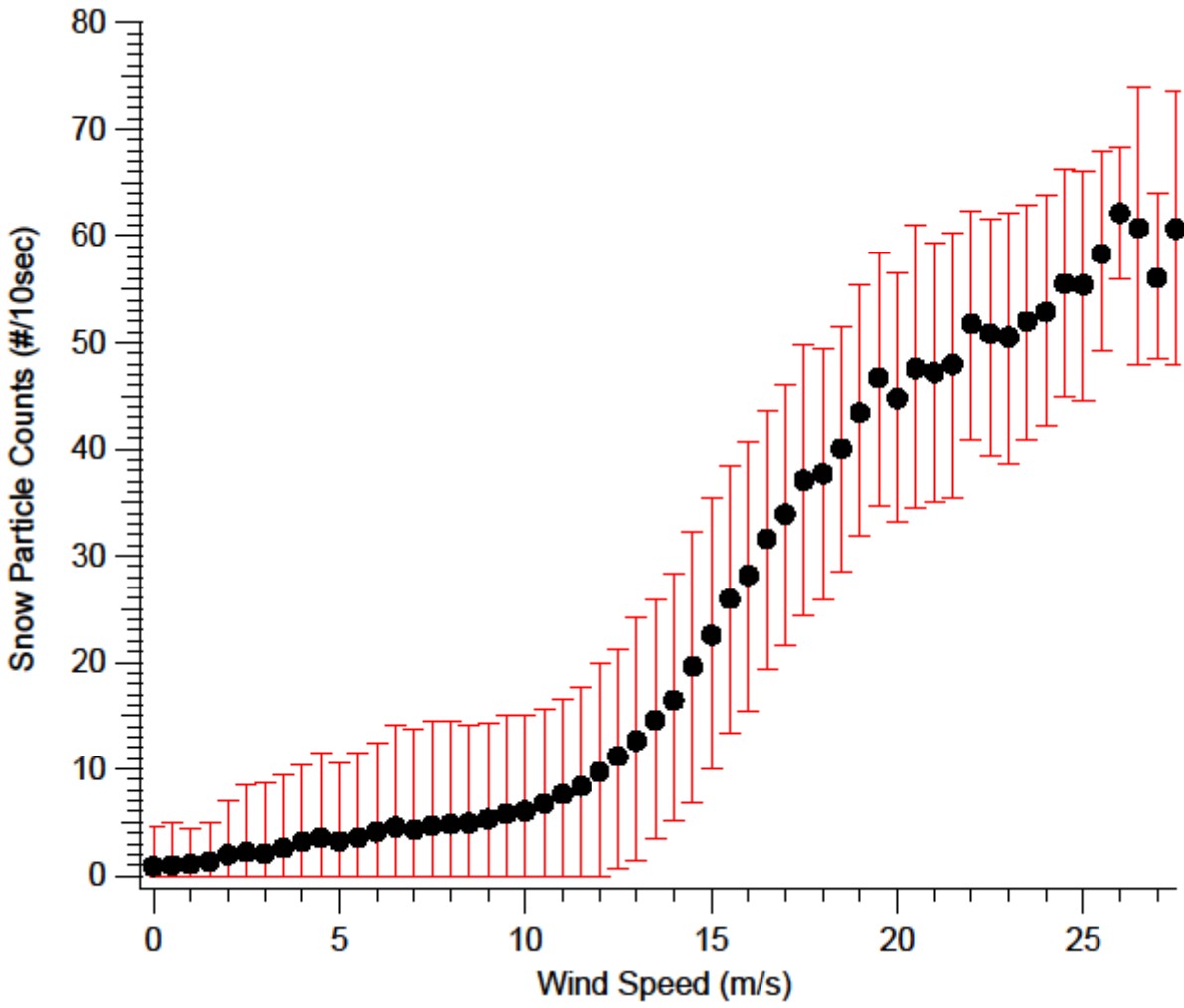

**Fig4 – Laser disdrometer snow particle number counts per 10 seconds with standard deviation, made at 5m above snow surface, as a function of wind speed.**

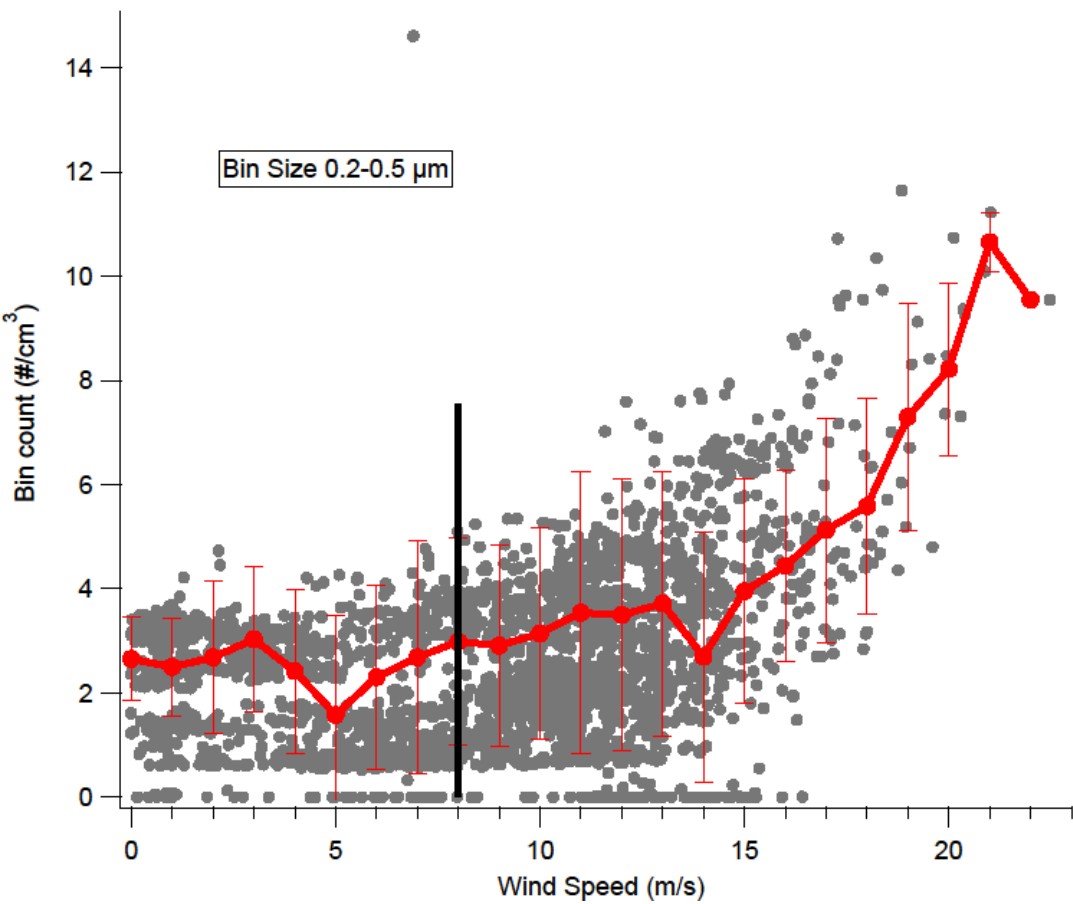

**Fig5 –** **Particle counts from the Lighthouse OPC in the 200-500nm size bin as a function of wind speed. Raw data (grey) and the averaged count and standard deviation for each wind speed in a 1ms$^{-1}$ bin shown in red. 8 ms$^{-1}$ is denoted with a black line.**

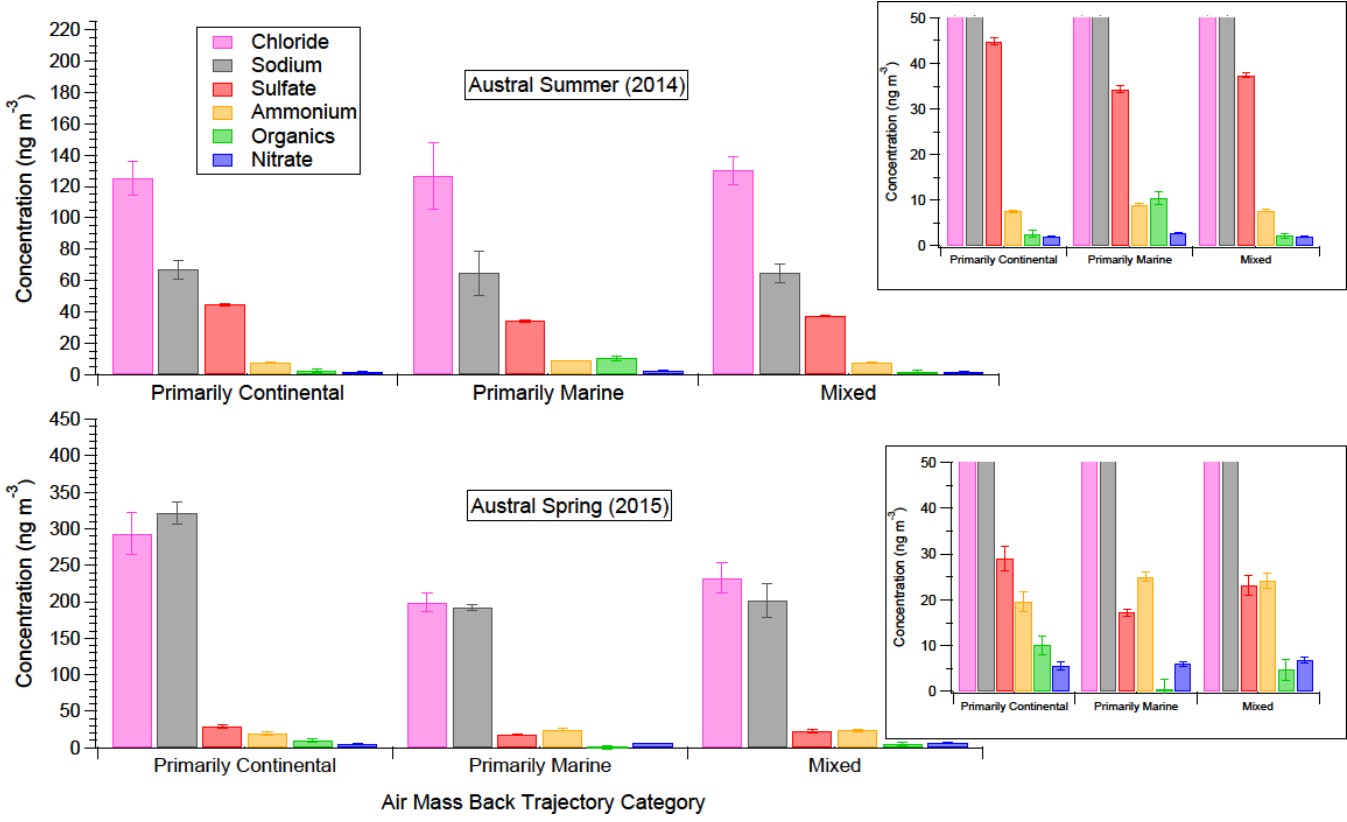

**Fig6 – Aerosol-phase concentrations of AMS species as a function of air mass back trajectory categories for the Austral summer (top) and spring (bottom) measurement durations, including inset zooms for lower concentration species. Error bars shown indicate standard error of the mean.**

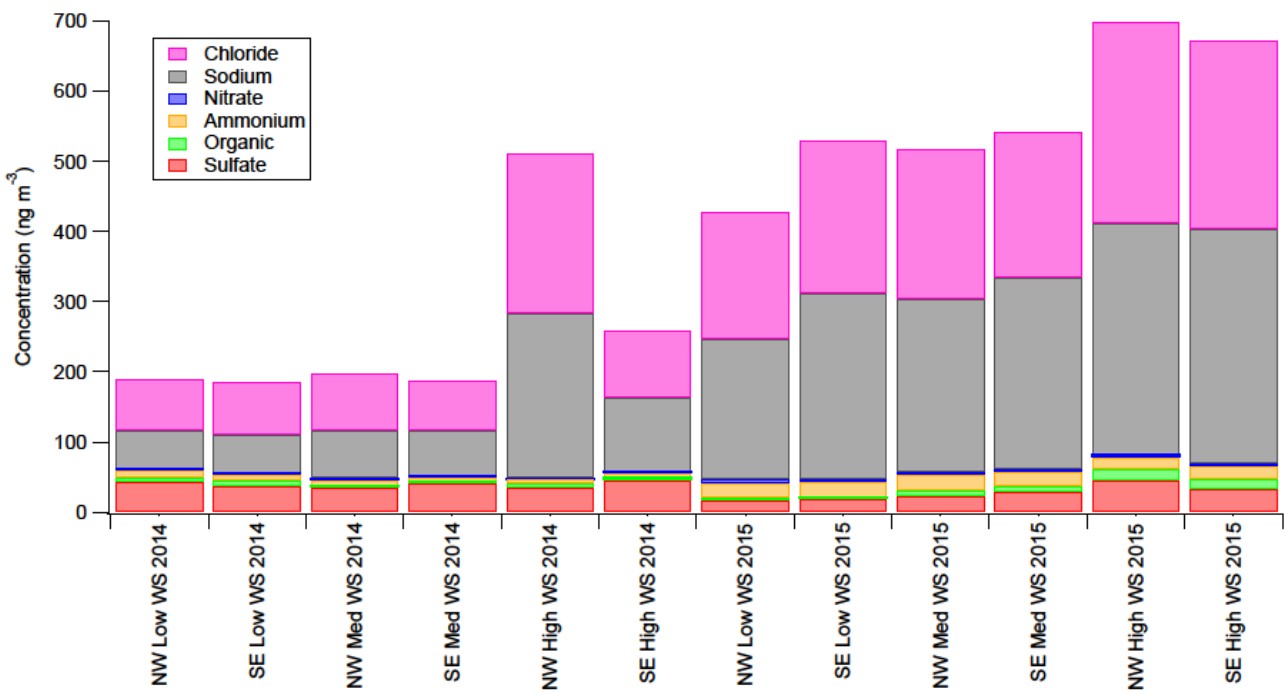

**Fig 7 – Aerosol-phase concentrations of AMS species as a function of low wind (0-1.99 ms$^{-1}$), medium wind (2-7.99 ms$^{-1}$), and high wind (8+ ms$^{-1}$) regimes for NW (marine) and SE (continental) wind directions. Error bars are omitted for readability but are: ± 10 for Chl, ± 10 for Na, ± 1 for Org, ± 3 for SO$_4$, ± 1 for NO$_3$, ± 2 for NH$_4$ for the 2014 spring/summer measurements and ±35 for Chl, ±35 for Na, ± 2 for Org, ±5 for SO$_4$, ± 2 for NO$_3$, and ± 3 for NH$_4$ for the 2015 winter/spring measurements.**

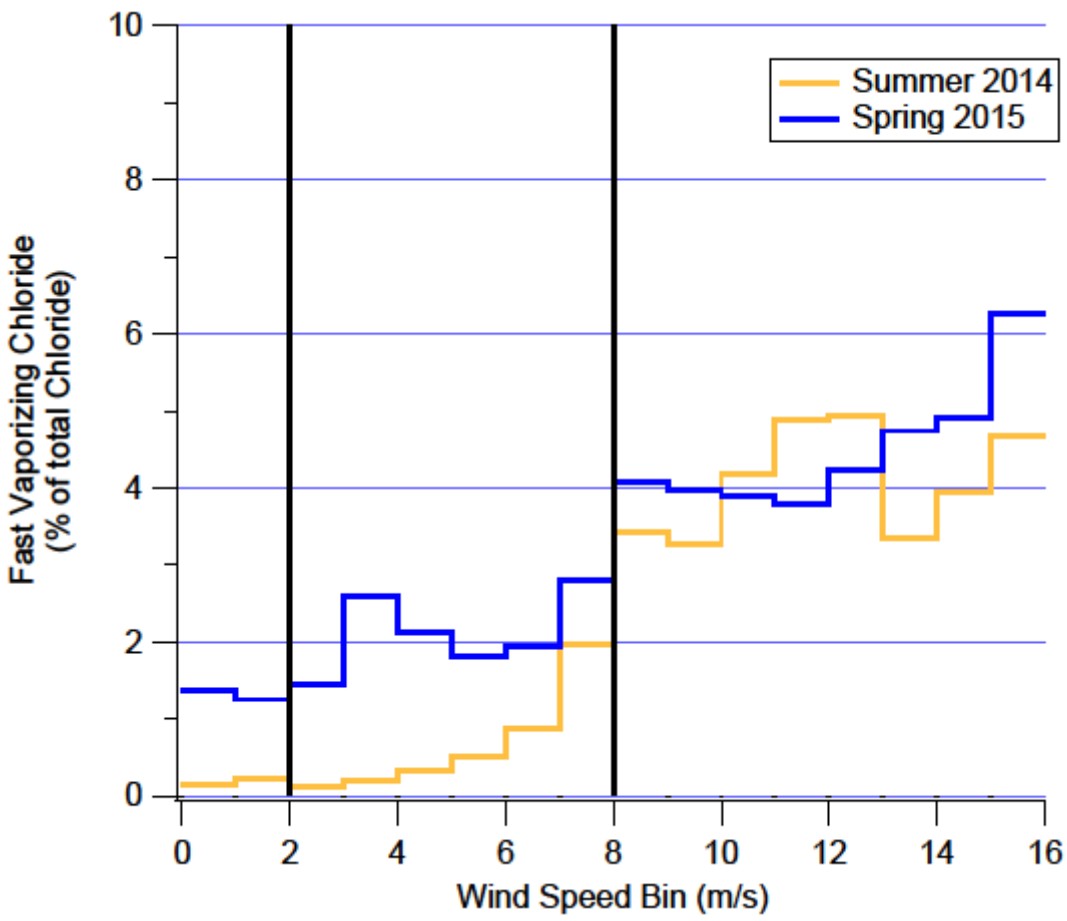

**Fig 8– Fast vaporizing chloride as a percent of total chloride measured (modelled) in the AMS as a function of wind speed for both the spring/summer field season (gold) and winter/spring field season(blue).**

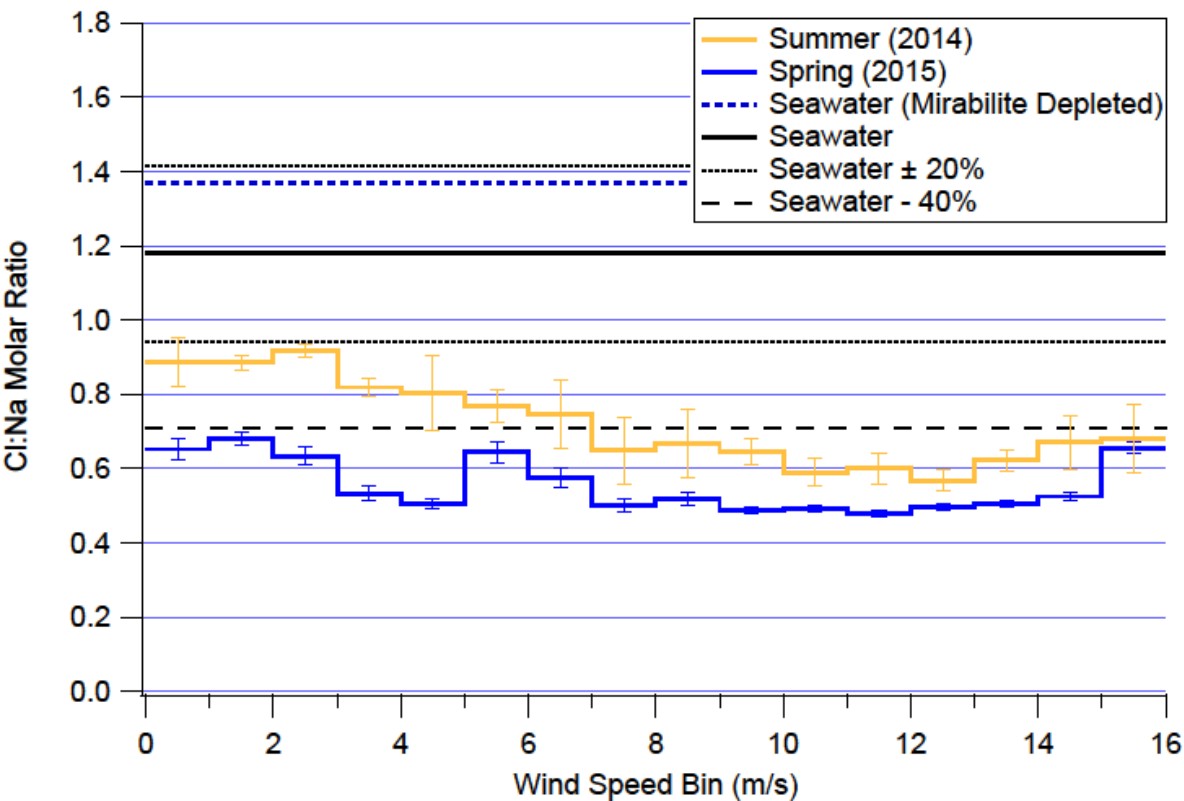

**Fig 9 – Cl:Na weight ratio as measured/modelled in the AMS as a function of wind speed for both overall field seasons. Error bars shown indicate standard error of the mean.**

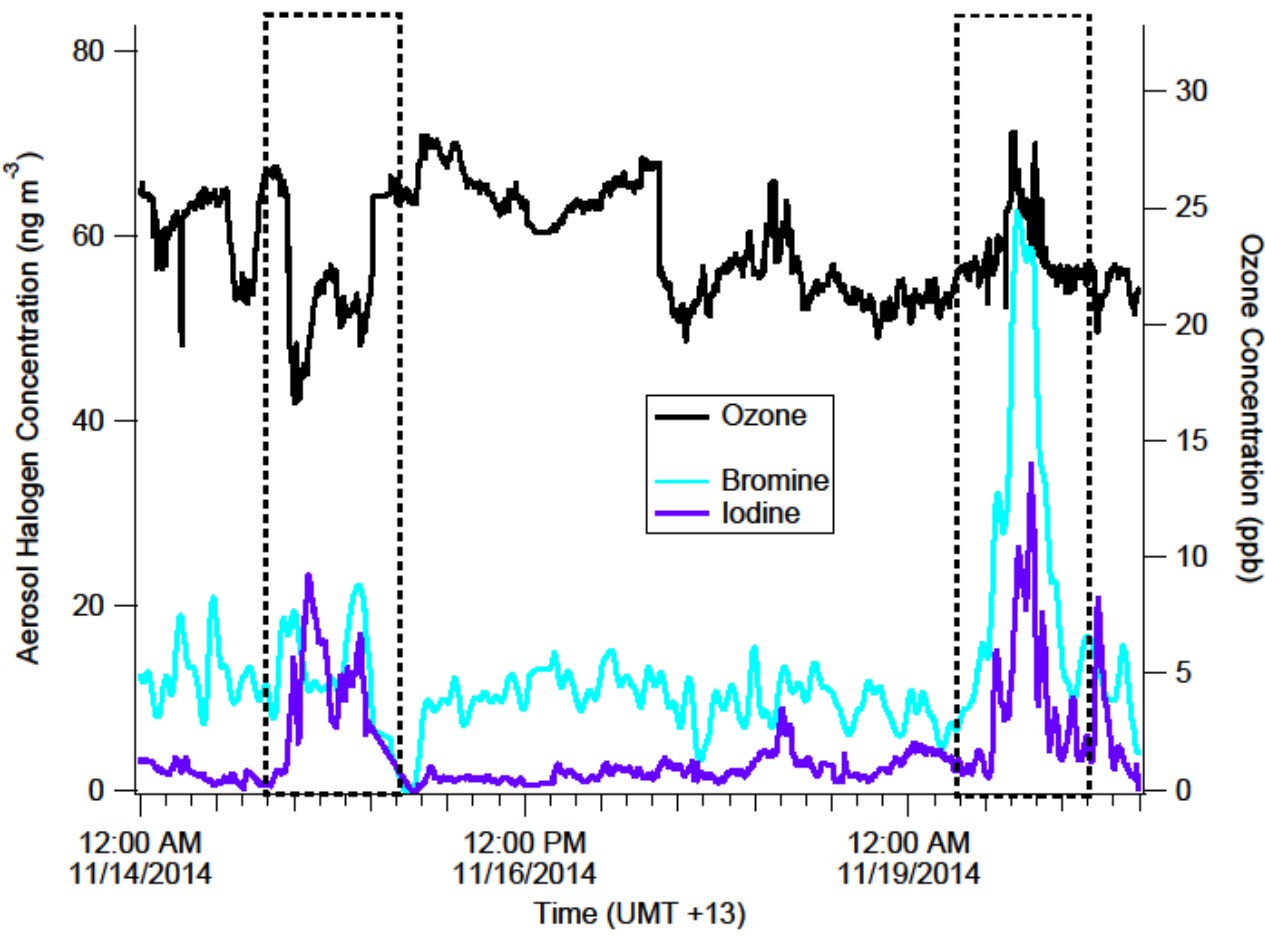

**Fig 10 – Time series of iodine (purple) and bromine (light blue) from the AMS as well as gas-phase ozone concentration (black). Nov. 15 is boxed to show an iodine-ozone anti-correlation; Nov. 19 to show an iodine-ozone correlation.**

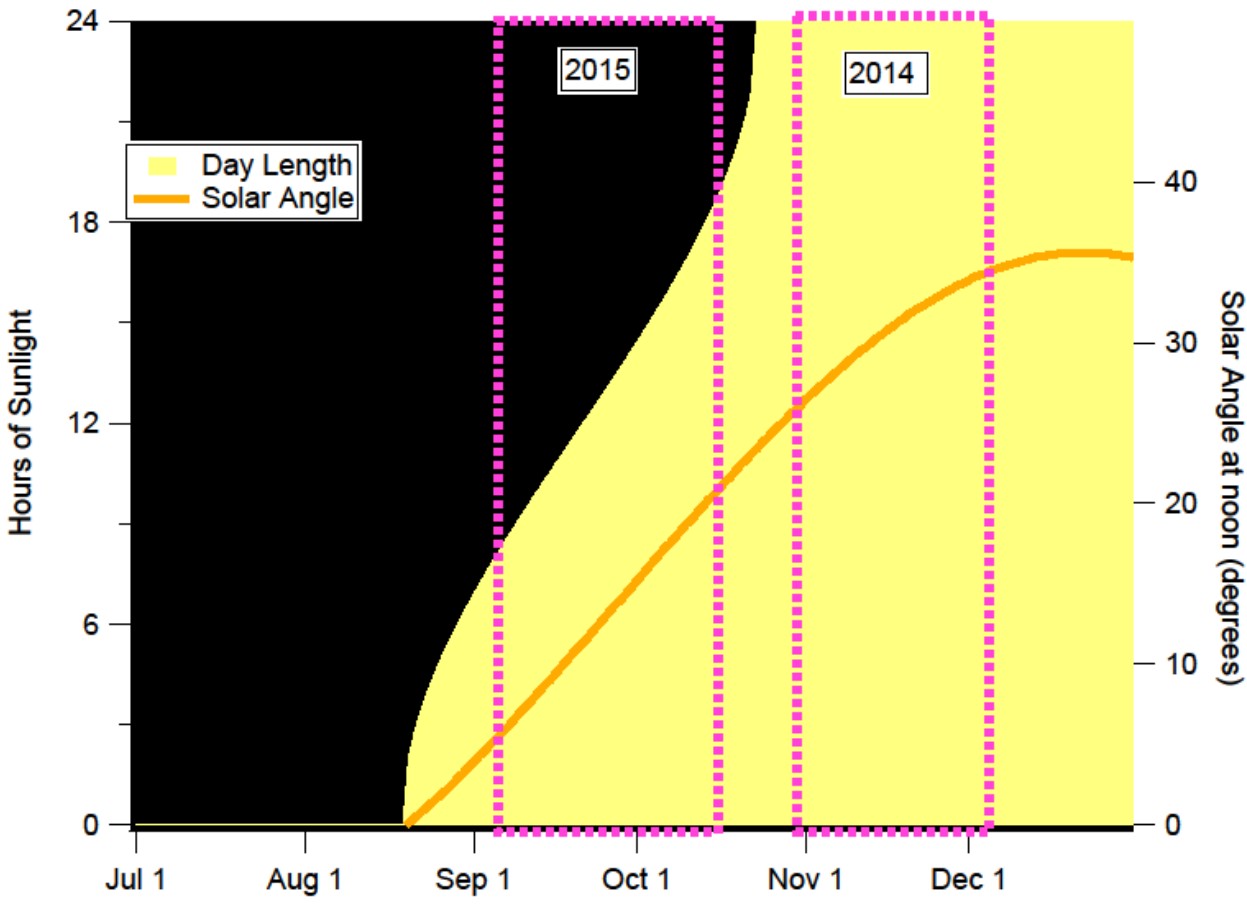

**Fig. S1 – Hours of sunlight per day and solar angle at noon for the field site near McMurdo Station, shown for the entire year. The durations of the two field campaigns are highlighted.**

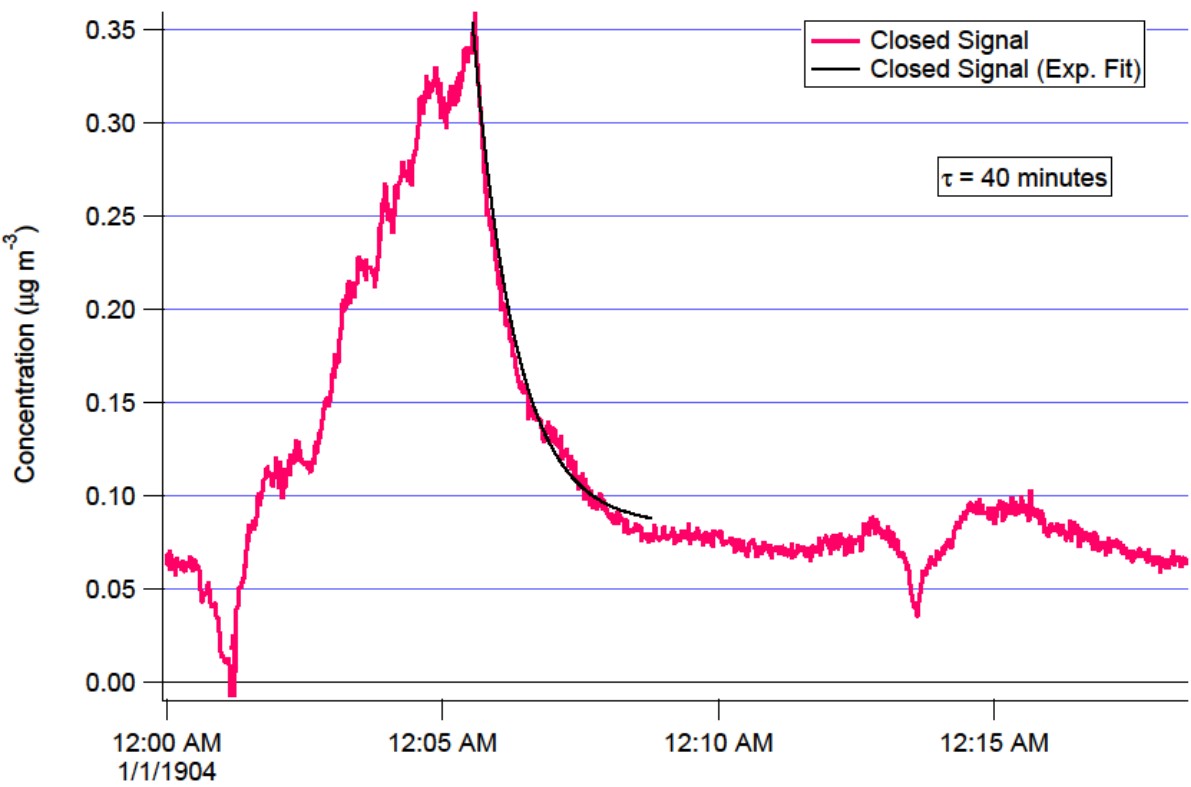

**Fig. S2 –** **Ambient data used in the vaporization model to calculate real ambient concentration from closed signal (pink) and exponential decay fit (black).**

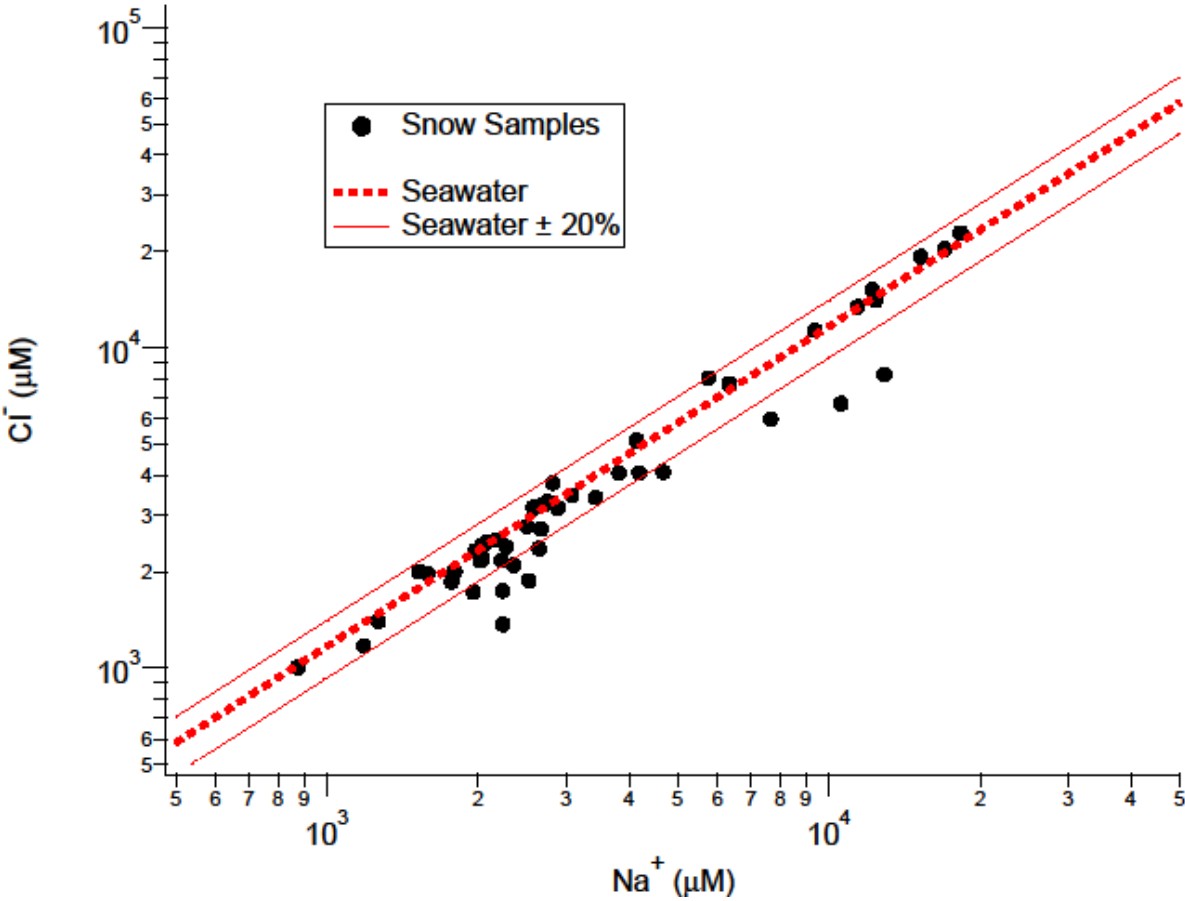

**Fig. S3 – Cl⁻ to Na⁺ ratio of snow samples from 2015 as well as the nominal seawater Cl⁻:Na⁺ ratio (red dotted lines).**

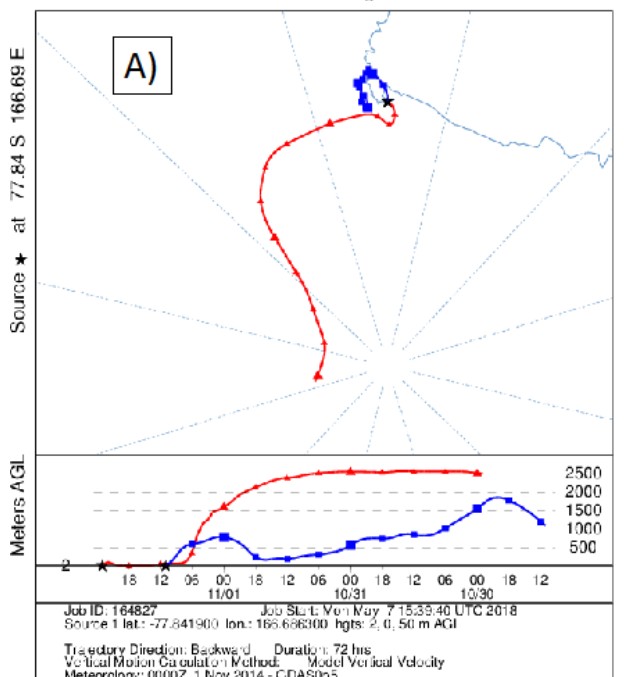 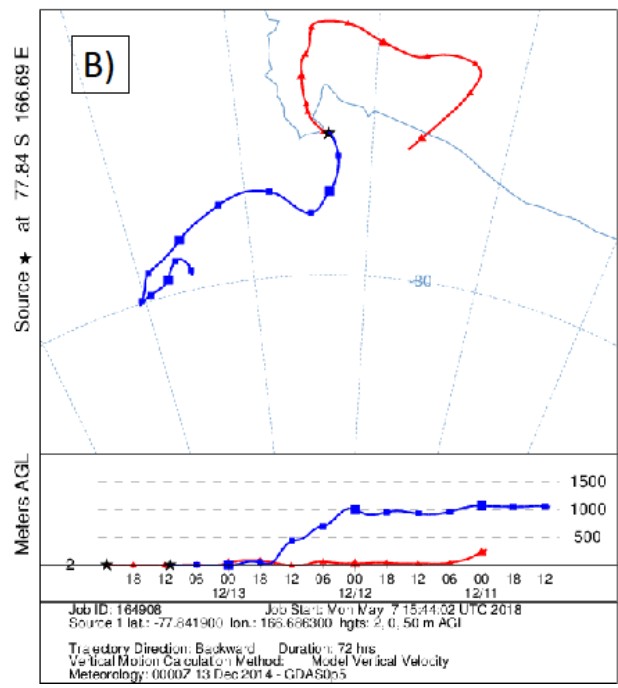

**Fig S4 – Examples of HYSPLIT back trajectories classified as Primarily Continental (red in A and blue in B), Primarily Marine (red in B) and Mixed (blue in A).**

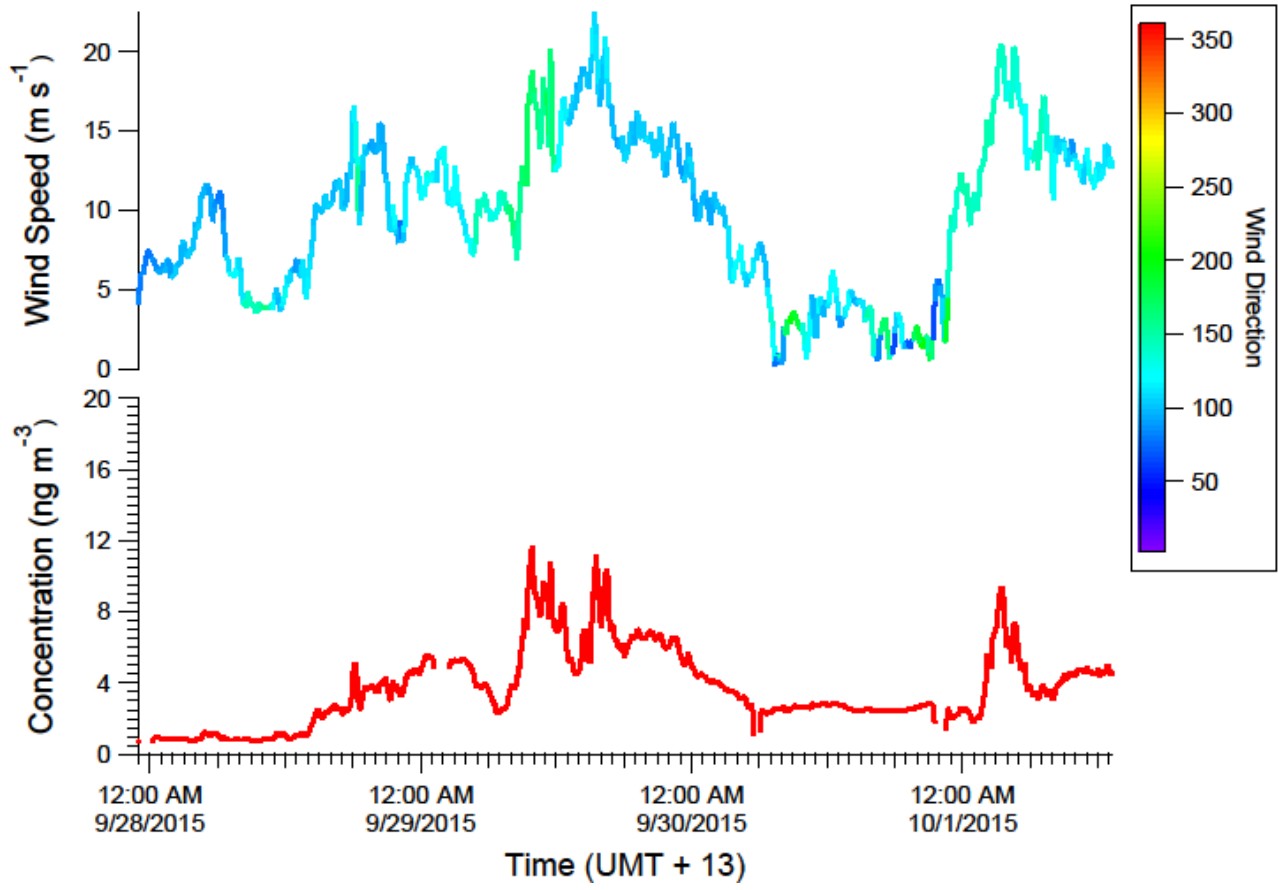

**Fig. S5 – Aerosol number (bottom) and wind speed (top) records as a function of time during a typical wind speed change.**

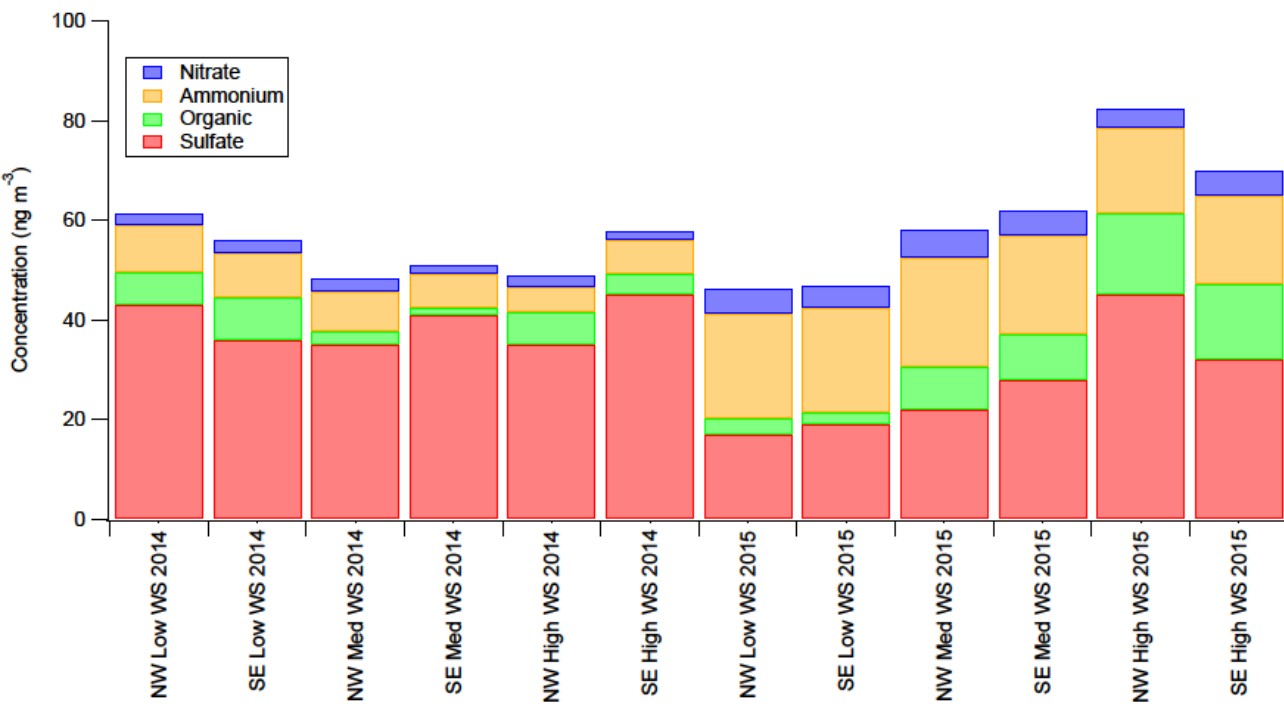

**Fig. S6 – As Figure 7 (including error bars), average AMS concentrations by wind regime without Na or Cl.**

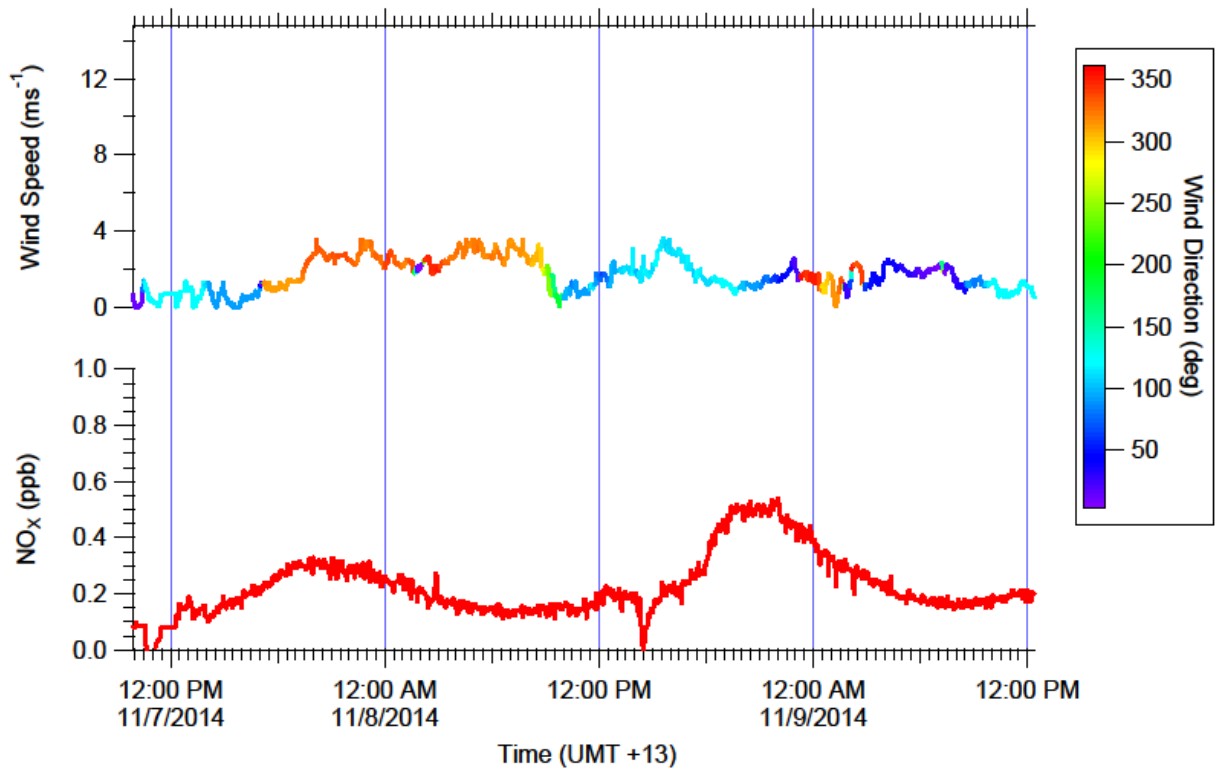

**Fig. S7 – NOₓ concentrations (bottom) and wind speed colored as a function of wind direction (top) measured during low wind conditions during the 2014 Spring/Summer campaign.**

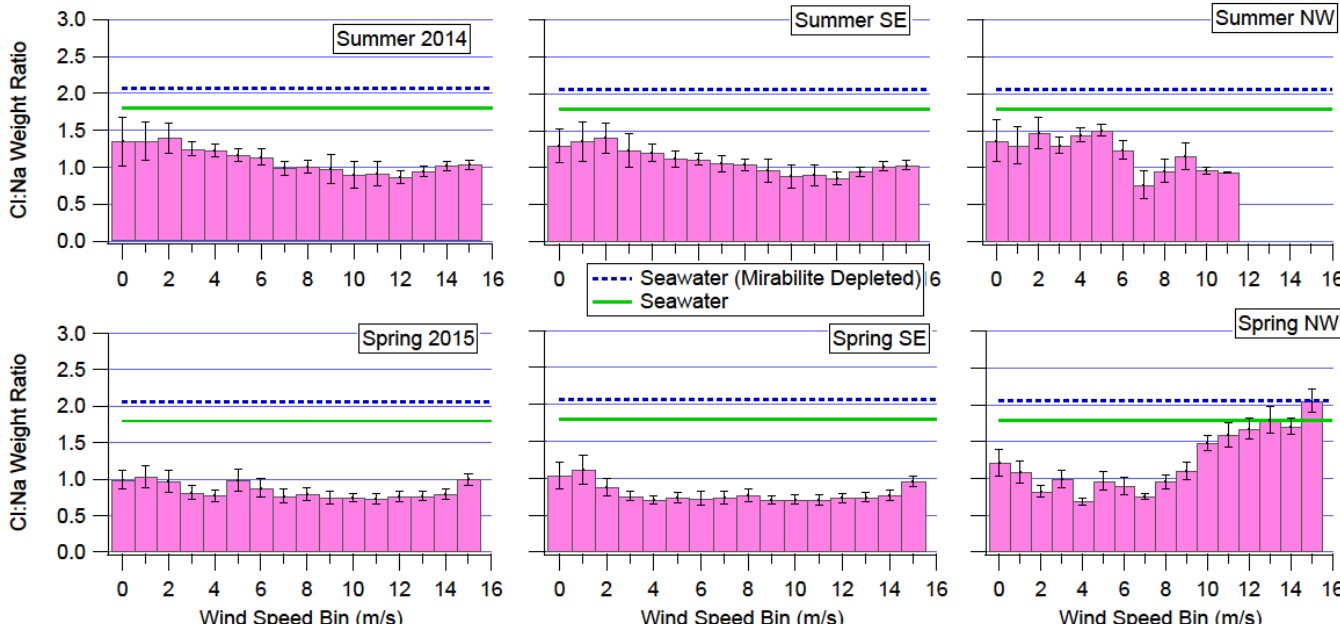

Fig. S8 – As Figure 9, Cl:Na weight ratio as measured/modelled in the AMS as a function of wind speed for both field seasons and parsed by wind direction: SE Wind (continental) in Summer (Middle Top), SE wind in spring (Middle Bottom), NW wind (marine) in summer (Right Top), and NW wind in spring (Right Bottom). Also shown are the seawater (green line) and seawater mirabilite-depleted (blue dashed line) Cl:Na weight ratios.

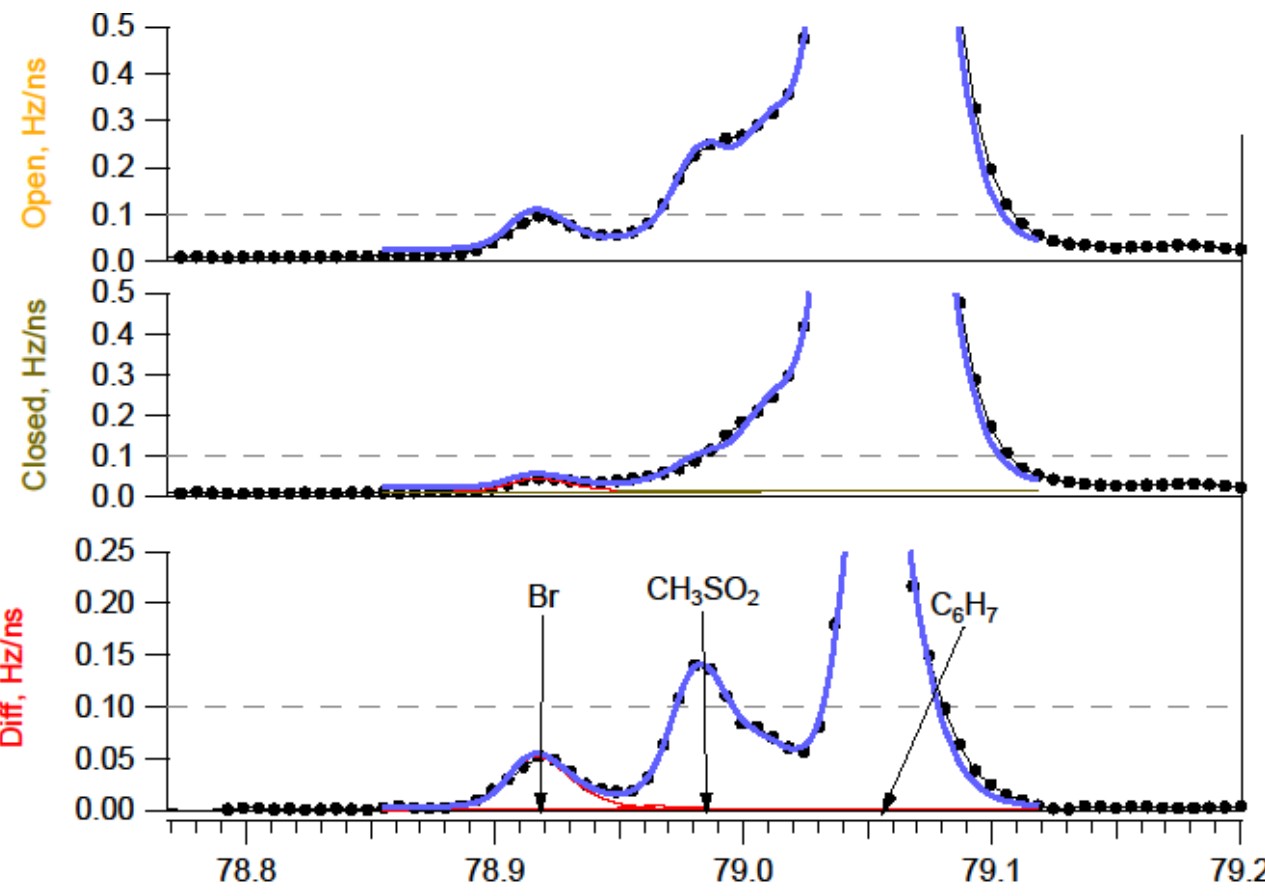

**Fig. S9 – Open, Closed, and Difference Spectra from High-Resolution fitting of AMS data at *m/z* 79. Data shown is averaged over the high Br event shown in Figure 10.**

| | Chl | | Chl St. Dev. | | Na$_{AMS}$ | | Na$_{AMS}$ St. Dev. | | Chl:Na$_{AMS}$ | | Chl:Na$_{AMS}$ Error | |
|---|---|---|---|---|---|---|---|---|---|---|---|---|
| | 2014 | 2015 | 2014 | 2015 | 2014 | 2015 | 2014 | 2015 | 2014 | 2015 | 2014 | 2015 |
| **Primarily Continental** | 125.12 | 293.37 | 10.94 | 28.86 | 66.99 | 321.28 | 5.99 | 15.01 | 1.87 | 0.91 | 0.23 | 0.09 |
| **Primarily Marine** | 126.81 | 199.13 | 21.30 | 13.21 | 64.78 | 192.26 | 14.00 | 4.04 | 1.96 | 1.04 | 0.54 | 0.07 |
| **Mixed** | 130.32 | 232.68 | 8.79 | 21.22 | 65.01 | 201.15 | 5.96 | 23.46 | 2.00 | 1.16 | 0.23 | 0.17 |

**Table S1 – Values for Chl and Na$_{AMS}$ and their standard deviations shown in Fig. 6 as well as the Chl:Na$_{AMS}$ ratio and its associated propagated error.**