# Peer review of "The importance of blowing snow to halogen-containing aerosol in coastal Antarctica: Influence of source region versus wind speed"

_Atmospheric Chemistry and Physics, 2018_

## Referee Comment (RC1) · Anonymous Referee #1 · 9 Jul 2018

Review of "The Importance of Blowing Snow to Antarctic Aerosols: Number Distribution and more than Source-Dependent Composition – results from the 2ODIAC campaign" by M. Giordano et al.

This study reports particle size and chemical composition of Antarctic aerosol particles with a focus on blowing snow and the influence of wind speed. Measurements were taken during two field studies in 2014 and 2015 covering Antarctic late winter to summer seasons. The location was near the McMurdo research station on the sea ice. The main findings of this work include that a) the aerosol chemical composition, in

particular sodium and chloride, is a function of wind speed rather than air mass origin as characterized through back trajectory analysis, b) the submicron particle halogen cycling is different from the supermicron particle, and c) more detailed studies are needed to fully understand the enrichment, depletion and cycling of halogen species in Antarctica. The manuscript is well-written and clearly organized. The manuscript can be published with minor revisions.

General comments

There seems to be a shift of focus between the title/introduction and conclusions. While the manuscript puts emphasis on the influence of wind speed on the aerosol population the conclusion mainly focuses on the halogen cycle. A section in the introduction on the halogen cycle is needed. For example, the two last result subsections use the reference Legrand et al. (2017) heavily, while he is not mentioned in the introduction.

The title is cryptic: "more than Source-Dependent Composition" is not a description of this work's content that will inform the reader directly of what this manuscript is about. What about: The importance of blowing snow to aerosol and the halogen cycle in coastal Antarctica: Influence of Source region versus wind speed.

The notation of the chemical species throughout the manuscript is confusing: e.g., chloride, Cl, Cl-, Chl. I suggest to introduce a notation that makes clear with which instrument the species was measured, e.g., Cl- for IC results, Chl for AMS results. Please check the manuscript throughout, sometimes the ionic charge is not provided (forgotten). Below are some hints, but I might not have spotted all.

Specific comments

Aerosol size distribution information based on AMS measurements are not provided in this manuscript. Therefore the description of size calibrations could be omitted to make the manuscript shorter. In section 2.3, p. 7, second paragraph: Please include a brief description of the model by Salcedo et al. (2010) so the reader does not have to look

up another publication to understand how the authors arrived from the e-folding time of 40 to the modelled Chl and Na concentrations as shown in Fig. 1. p. 9, l. 21: It is not clear whether those data points are shown as well. Please make this evident in the text.

p. 9, l. 27: "chloride losses" Do they refer as relative to Na+ or to the total chloride concentration? p. 10., l. 3: How can winter blizzard conditions be more consistent with the 2ODIAC campaign which happened later in the season?

p. 12, l. 30: Are those the total numbers of trajectories or just at one release height? Please specify in the manuscript.

Figure 5: It would be helpful to make all data points grey and overlay them by wind speed binned averages with standard deviations.

p. 14, l. 15f: What was the size range of particles that Hara et al. (2014) measured? From the text it is not evident that the comparison makes sense.

Section 3.7: Could the vertical displacement of the back trajectories be an indicator of possible processes? The authors looked at source regions which do not provide a clear hint, but potential influence of high tropospheric or stratospheric air masses could be important.

Conclusions: p. 22, l. 8: The mixing state of particles has not been mentioned before in the manuscript. If there is a strong argument for external halogen mixture, this needs to be included in the respective results section. It will also be highly informative to include a couple of sentences what more information of the halogen cycle in Antarctica will be important for.

Technical comments

p. 3, l. 10: elaborate which elements are meant by "marine elements"

p. 5, l. 26: "an" instead of "An"

p. 7, l. 11: "Figure 1 shows how the wind direction and..." The wind direction is not shown, but can be added easily.

p. 7, l. 28: "and sodium my occur in the data."

p. 8, l. 1: "freezer in Antarctica." Admitted, there is a limited total number of freezers in Antarctica, but more specificity is desirable, e.g., the research site's freezer.

p. 8, l. 19: delete "while in the field and subsequent data analysis"

p. 8, l. 21f: It is not clear what is meant by "increasing temporal resolution patterns."

p. 9, l. 6: "it"

p. 9, l. 27: "was" instead of "is not clear."

p. 10, l. 24 / p. 11, l. 10-12: Do you mean "SO4 2-"?

p. 11, l. 2f: The abbreviations have not been introduced.

p. 11, l. 32: sulphate or sulfate?

p. 12, l. 15: "differences that"

p. 12, l. 29: SI or Supplemental Information?

p. 13, l. 19: "Two" instead of "Several"

p. 13, l. 25: "which in turn is a function..."

p. 14, l. 14: "These results are..."

p. 14, l. 33: It is unclear what is meant by "make up the aerosol to inland continental snow."

p. 15, l. 27: It is unclear what is meant by " by extension concentrations."

p. 16, l. 9f and elsewhere: The hyphens turn out very long and spaces are missing.

p. 17, l. 2: "region" instead of "point". The sources are rather regions that specific points.

p. 17, l. 5: Delete "it was quickly noted that"

p. 17, l. 16: replace "in the AMS" by "as measured by the AMS". The original formulation, here and elsewhere, sound like the species are generated within the AMS.

p. 17, l. 33 and elsewhere: sometimes it is written Figure x or Fig. x or figure. Check the journal style.

p. 18, l. 5: " that the increased chloride concentrations as measured by the AMS are not..."

p. 18, l. 13 and elsewhere: Do you refer to the IC measured species Cl- and Na+? Or the AMS derived species Chl and Na?

p. 19, l. 13 and elsewhere: if this is the AMS derived ratio use Chl:Na

p. 19, l. 25: Do you mean at intermediate high wind speeds?

p. 20, l. 14: "with the AMS" instead of "in the AMS"

p. 20, l. 20: "by Maffezzoli" according to the sentence structure

p. 21, l. 21: "summer", l. 26: delete "overall"

p. 21, l. 27f: the notation of the chemical species is not clear,

p. 21, l. 29: Based on AMS measurements

---

## Referee Comment (RC2) · Anonymous Referee #2 · 29 Jul 2018

**Comments on the manuscript titled "The importance of Blowing snow to Antarctic Aerosol: Number distribution and more than Source-Dependent Composition –results from the 2ODIAC campaign" by Giordano et al.**

This manuscript reports a 2-season (austral spring-summer 2014 and winter-spring 2015) campaign near McMurdo station, Antarctica, with research focuses on aerosol number density and composition as a function of air mass origin (via back trajectory analysis) and local meteorological conditions (mainly wind speeds). Their major conclusions are that 1) blowing snow has significant impacts on aerosol counts and chemical compounds (e.g. Cl:Na ratio, Br- and I- concentrations); 2) air mass origin has little influence. It is a well-written manuscript with in depth discussions in each section. I should be published in ACP with minor revisions.

General comments:
The first part of the Introduction section contains an overview of Antarctic aerosol studies, but mainly in instrumental aspect. Given the title/research focus of this manuscript is about blowing snow and relevant aerosol, a brief introduction is needed. For example, on page 3 lines 27-29, it reads 'Some studies in Antarctica have hypothesized that elevated wind speeds can cause sea salt concentration differences through unknown mechanisms', then which studies do you refer to? References should be supplied here. Yang et al. (2008) has proposed a mechanism of SSA production from blowing snow through sublimating saline wind-blown snow particles, which parameterisation has been implemented in global models to investigate high latitude SSA (e.g. Levine et al., 2014; Huang and Jaegle 2016; Rhode et al., 2017). Frost flowers are also thought to be a SSA source. More information can be found in e.g. Abbatt et al., (2012).
Yang, et al.: Sea salt aerosol production and bromine release: Role of snow on sea ice, Geophys. Res. Lett., 35 (L16815), doi:10.1029/2008gl034536, 2008.
Huang J., Jaeglé, L., Wintertime enhancements of sea salt aerosol in polar regions consistent with a sea-ice source from blowing snow, ACP, doi:10.5194/acp-2016-972 (2016).
Levine, et al.: Sea salt as an ice core proxy for past sea ice extent: A process-based model study, J. Geophys. Res. Atmos., 119, 5737–5756, doi:10.1002/2013JD020925 (2014).
Rhodes, et al.: Sea ice as a source of sea salt aerosol to Greenland ice cores: a model-based study, Atmos. Chem. Phys., 17, 9417-9433, https://doi.org/10.5194/acp-17-9417-2017 (2017).
Abbatt, et al: Halogen activation via interactions with environmental ice and snow in the polar lower troposphere and other regions, Atmos. Chem. Phys., 12, 6237-6271, https://doi.org/10.5194/acp-12-6237-2012, 2012.

Section 3.7 (about bromine): Sander et al.'s (2003) global sea spray dataset clearly shows a general bromine depletion in micron mode, however, in ultra-fine mode, bromine is largely enriched. A similar phenomenon is also seen in polar dataset (Legrand et al., 2016). Do you have any comments on this size-dependent behaviour and relevance to your data interpretation?
Sander, et al.: Inorganic bromine compounds in the marine boundary layer: A critical review, *Atmos. Chem. Phys.*, **3**, 1301–1336, 2003.

Specific comments:
P5L26: 'An' should be 'an'
P6L1-2: '50m' or '50mL'? A full stop is needed before the second '50 mL' (?)
P7L28: delete the duplicated word 'in'

SI Fig.S3: remove duplicated brackets in the caption.
SI Fig. S4: where are 'squares'? which 'solid' line (red or coloured) do you refer to?

---

## Author Comment (AC1) · 1 Oct 2018

Response to Reviewer #1: The authors thank Reviewer #1 for their thoughtful and constructive comments on the manuscript. The reviewer's comments have been implemented as below:

Reviewer - General Comments: There seems to be a shift of focus between the title/introduction and conclusions. While the manuscript puts emphasis on the influence of wind speed on the aerosol population the conclusion mainly focuses on the halogen

cycle. A section in the introduction on the halogen cycle is needed. For example, the two last result subsections use the reference Legrand et al. (2017) heavily, while he is not mentioned in the introduction.

Authors - A section on the halogen cycle as it relates to Antarctic aerosols has been added to the introduction:

"Halogen-containing aerosols in the Antarctic are connected to the depletion of near-surface tropospheric ozone, a greenhouse gas, (Kalnajs et al., 2013) and may also contribution to new particle formation (Sipilä et al., 2016). Numerous remote sensing measurements of halogen oxides in the boundary layer (Saiz-Lopez et al., 2007; Simpson et al., 2015; Prados-Roman et al., 2018) show active bromine and iodine chemistry in the Antarctic troposphere. Chemical ionization mass spectrometry measurements of Br2, BrCl, and BrO confirmed the coupling of bromine and chlorine chemistry in the springtime Antarctic troposphere (Buys et al., 2013). Reactions with non-methane hydrocarbons (e.g. Ramacher et al., 1999), elemental mercury (Saiz-Lopez et al., 2008), and DMS (von Glasow et al., 2004; Read et al., 2008) can all have large impacts on tropospheric composition over the poles. Additionally, some work has also been done examining iodine oxides' role in forming ultrafine aerosols (O'Dowd et al., 1998; McFiggans et al., 2004; Saunders and Plane 2005). Despite significant progress in our understanding of atmospheric halogen gases, a lack of observational data capturing the interactions between gas-phase halogens, aerosols, and snow persists, limiting our understanding of these multiphase processes. "

Reviewer - The title is cryptic: "more than Source-Dependent Composition" is not a description of this work's content that will inform the reader directly of what this manuscript is about. What about: The importance of blowing snow to aerosol and the halogen cycle in coastal Antarctica: Influence of Source region versus wind speed.

Authors - The authors agree with the reviewer and have changed the title to incorporate the suggestion of the reviewer: new title "The importance of blowing snow to halogencontaining aerosol in coastal Antarctica: Influence of source region versus wind speed".

Reviewer - The notation of the chemical species throughout the manuscript is confusing: e.g., chloride, Cl, Cl-, Chl. I suggest to introduce a notation that makes clear with which instrument the species was measured, e.g., Cl- for IC results, Chl for AMS results. Please check the manuscript throughout, sometimes the ionic charge is not provided (forgotten). Below are some hints, but I might not have spotted all.

Authors - The authors thank the reviewer for pointing out that the nomenclature is not 100% consistent. A sentence has been added at the end of Section 3 denoting which notation implies which measurement. A few parenthetical reminders have also been added throughout the rest of the text due to the length of the manuscript.

"In this manuscript, 3 notations are presented for chemical identities and correspond to either instrument used to quantify or to their presence regardless of molecular structure. 1) For any ion measured with the IC system, ionic charge is noted (e.g. Cl-, Na+). 2) For compounds measured with the AMS, notation in the text follows AMS standard notation (e.g. Chl for chloride, SO4 for sulfate, NO3 for nitrate) with the exception of sodium measured by the AMS and corrected through the process described in S2.3 – NaAMS. 3) For general and/or hypothetical discussions without implication to measurement or molecular compounding, elements are referred to simply by their periodic table notation or written out (e.g. chloride, Cl, sodium, Na)."

Authors - Some exceptions are left in the text (e.g. "chloride salts" in S2.3) to avoid having to add another notation (e.g. XCl) but the meaning in these few cases should be obvious for all readers.

Specific Comments

Reviewer - Aerosol size distribution information based on AMS measurements are not provided in this manuscript. Therefore the description of size calibrations could be omitted to make the manuscript shorter.

Authors - The sentences describing the AMS calibrations have been replaced with: "The AMS was calibrated as described in Giordano et al., 2017"

Reviewer - In section 2.3, p. 7, second paragraph: Please include a brief description of the model by Salcedo et al. (2010) so the reader does not have to look up another publication to understand how the authors arrived from the e-folding time of 40 to the modelled Chl and Na concentrations as shown in Fig. 1. p. 9, l. 21: It is not clear whether those data points are shown as well. Please make this evident in the text.

Authors - A brief description of the Salcedo model has been added to the text: "The model employed by Salcedo et al. (2010) assumes an incoming mass flux of a slow-vaporizing species which impacts the vaporizer, sticks to it, and then desorbs at a rate proportional to the amount on the vaporizer at any given time. This assumption results in an explicit ordinary differential equation."

A figure showing the data fit to obtain the e-folding time has been added to the supplement for this work (SI Fig. 2).

Reviewer - p. 9, l. 27: "chloride losses" Do they refer as relative to Na+ or to the total chloride concentration?

Authors - The text has been changed to reflect the cited manuscript: "Virkkula et al. (2006) noted chloride losses (relative to Na+) linked with. . ."

Reviewer - p. 10., l. 3: How can winter blizzard conditions be more consistent with the 2ODIAC campaign which happened later in the season?

Authors - The text states "blizzard and strong wind conditions", where "strong wind conditions" is consistent with what was measured in the Spring (2015) campaign (over 60% of the total time of that deployment seeing winds over 8ms-1). "Blizzard" is left in because that is part of how Hara et al., 2004 describes their results.

Reviewer - p. 12, l. 30: Are those the total numbers of trajectories or just at one release height? Please specify in the manuscript.

Authors - A parenthetical addition has been made to make the sentence clearer: "(37 and 36 days, respectively, at 3 hour intervals for each release height)".

Reviewer - Figure 5: It would be helpful to make all data points grey and overlay them by wind speed binned averages with standard deviations.

Authors - The figure has been modified as recommended.

Reviewer - p. 14, l. 15f: What was the size range of particles that Hara et al. (2014) measured? From the text it is not evident that the comparison makes sense.

Authors - "primarily in the 500nm - 2$\mu$m size ranges" has been added to the text. This is on the upper range of what would be captured by the AMS but should be well encompassed by the Lighthouse OPC used.

Reviewer - Section 3.7: Could the vertical displacement of the back trajectories be an indicator of possible processes? The authors looked at source regions which do not provide a clear hint, but potential influence of high tropospheric or stratospheric air masses could be important.

Authors - The HYSPLIT trajectories examined often showed a lot of movement altitudinally over their lifetimes. However, analyzing how the vertical mobility of an air mass over its lifetime affects the halogen and/or aerosol content is outside of the scope of this manuscript. However, the reviewer brings up an excellent point. The following has been added to the text in S3.7 to make the reviewer's point known and to point out potential future work: "The results presented here suggest that source regions for air masses do not significantly affect aerosol halogen concentrations in Antarctica. However, we do not here assess what, if any, impact that vertical displacement during transport may have on the halogen concentrations in the aerosol phase. Due to the low concentrations of aerosol-phase halogens measured, the uncertainty in the HYSPLIT runs, and the fact that the air masses examined in most of the HYSPLIT runs had large altitude changes during their back trajectory lifetimes, assessing the effect of vertical

displacement of an air mass on halogen or aerosol concentrations or compositions is outside of the scope of this manuscript. However, it should be noted that models have shown there to be significant differences in the vertical distribution of gas-phase halogens in the boundary layer (Lehrer et al., 2004; Saiz-Lopez et al., 2008). Any vertical movement of an air mass could therefore have significant impacts on the halogen cycle and halogen-aerosol interactions."

Reviewer - Conclusions: p. 22, l. 8: The mixing state of particles has not been mentioned before in the manuscript. If there is a strong argument for external halogen mixture, this needs to be included in the respective results section. It will also be highly informative to include a couple of sentences what more information of the halogen cycle in Antarctica will be important for.

Authors - The statement about mixing state has been removed since no evidence for it has been presented in the text. The following has been added to demonstrate the importance of understanding the halogen cycle: "A clearer understanding of the halogen cycle over Antarctica will result in better modelling capabilities of the oxidative capacity of the polar troposphere which impacts both current climate (e.g. DMS oxidation) and paleoclimate reconstructive abilities."

Technical Comments:

Reviewer - (Grammatical/Spelling comments:) p. 5, l. 26: "an" instead of "An" p. 7, l. 28: "and sodium my occur in the data." p. 8, l. 19: delete "while in the field and subsequent data analysis" p. 9, l. 6: "it" p. 9, l. 27: "was" instead of "is not clear." p. 12, l. 15: "differences that" p. 13, l. 19: "Two" instead of "Several" p. 13, l. 25: "which in turn is a function: : :" p. 14, l. 14: "These results are: : :" p. 16, l. 9f and elsewhere: The hyphens turn out very long and spaces are missing. p. 17, l. 2: "region" instead of "point". The sources are rather regions that specific points. p. 17, l. 5: Delete "it was quickly noted that" p. 18, l. 5: " that the increased chloride concentrations as measured by the AMS are not: : :" p. 19, l. 13 and elsewhere: if this is the AMS derived ratio use

[Figure]

Chl:Na p. 20, l. 14: "with the AMS" instead of "in the AMS" p. 20, l. 20: "by Maffezzoli" according to the sentence structure p. 21, l. 21: "summer", l. 26: delete "overall" p. 21, l. 29: Based on AMS measurements

Authors - The above grammatical/spelling comments have all been implemented into the manuscript.

Reviewer - p. 17, l. 33 and elsewhere: sometimes it is written Figure x or Fig. x or figure. Check the journal style.

Authors - All instances of "Figure" have been changed to "Fig." unless they are at the start of a sentence for consistency.

Reviewer - p. 17, l. 16: replace "in the AMS" by "as measured by the AMS". The original formulation, here and elsewhere, sound like the species are generated within the AMS.

Authors - There are a few instances where "in the AMS" is the correct usage in context (e.g. S2.3, other occurrences regarding vaporization) but the rest of the occurrences have been changed as suggested.

Reviewer - p. 3, l. 10: elaborate which elements are meant by "marine elements"

Authors - "(Li, K, Mg, Ca, Sr)" has been added to the text to represent what was measured (minus Na since it is a focus here) by Weller et al., 2008.

Reviewer - p. 7, l. 11: "Figure 1 shows how the wind direction and: : :" The wind direction is not shown, but can be added easily.

Authors - ". . .direction and. . ." referred to an older version of the figure and has been removed. Wind direction changes are not relevant to the decay period shown here.

Reviewer - p. 8, l. 1: "freezer in Antarctica." Admitted, there is a limited total number of freezers in Antarctica, but more specificity is desirable, e.g., the research site's freezer.

Authors - "in the Crary Lab" has been added to the text.

Reviewer - p. 8, l. 21f: It is not clear what is meant by "increasing temporal resolution patterns."

Authors - The sentence has been removed because it does not add anything to the context of the rest of the paragraph.

Reviewer - p. 10, l. 24 / p. 11, l. 10-12: Do you mean "SO4 2-"?

Authors - The authors thank the reviewer for this catch; the instances on p.10 and p.11 have been corrected to "2-"

Reviewer - p. 11, l. 2f: The abbreviations have not been introduced.

Authors - The abbreviations have been removed and the sentence changed to read: "This is not in agreement with results from Eom et al. (2016) which measured higher inorganic salt concentrations in summer aerosols by Raman microspectrometry and attenuated total reflection Fourier transform infrared imaging techniques"

Reviewer - p. 11, l. 32: sulphate or sulfate?

Authors - "Sulphate" has been changed to "sulfate" to be consistent with the rest of the manuscript and abide by IUPAC recommendations.

Reviewer - p. 12, l. 29: SI or Supplemental Information?

Authors - "Supplemental Information" has been changed to "SI" for consistency throughout the manuscript.

Reviewer - p. 14, l. 33: It is unclear what is meant by "make up the aerosol to inland continental snow."

Authors - The sentence has been changed to make the meaning clearer: ". . .may explain longer range transport of chemical species (found primarily in the aerosol phase) to inland continental snow."

Reviewer - p. 15, l. 27: It is unclear what is meant by " by extension concentrations."

Authors - The sentence has been changed to make the meaning clearer: "...and, by extension, aerosol concentrations."

Reviewer - p. 18, l. 13 and elsewhere: Do you refer to the IC measured species Cl- and Na+? Or the AMS derived species Chl and Na?

Authors - This comment is addressed in the "General Comments" section above.

Reviewer - p. 19, l. 25: Do you mean at intermediate high wind speeds?

Authors - "than at high wind speeds" has been removed to make the sentence clearer. Sentence now reads: "This is near the same wind speed at which snow particle numbers increase more rapidly (7-13ms-1; Fig. 4)."

Reviewer - p. 21, l. 27f: the notation of the chemical species is not clear,

Authors - This comment is addressed in the "General Comments" section above.

  Response to Reviewer #2

General Comments

Reviewer - The first part of the Introduction section contains an overview of Antarctic aerosol studies, but mainly in instrumental aspect. Given the title/research focus of this manuscript is about blowing snow and relevant aerosol, a brief introduction is needed. For example, on page 3 lines 27-29, it reads 'Some studies in Antarctica have hypothesized that elevated wind speeds can cause sea salt concentration differences through unknown mechanisms', then which studies do you refer to? References should be supplied here. Yang et al. (2008) has proposed a mechanism of SSA production from blowing snow through sublimating saline wind-blown snow particles, which parameterisation has been implemented in global models to investigate high latitude SSA (e.g. Levine et al., 2014; Huang and Jaegle 2016; Rhode et al., 2017). Frost flowers are also thought to be a SSA source. More information can be found in e.g. Abbatt et al., (2012). Yang,

et al.: Sea salt aerosol production and bromine release: Role of snow on sea ice, Geophys. Res. Lett., 35 (L16815), doi:10.1029/2008gl034536, 2008. Huang J., Jaeglé, L., Wintertime enhancements of sea salt aerosol in polar regions consistent with a sea-ice source from blowing snow, ACP, doi:10.5194/acp-2016-972 (2016). Levine, et al.: Sea salt as an ice core proxy for past sea ice extent: A process-based model study, J. Geophys. Res. Atmos., 119, 5737–5756, doi:10.1002/2013JD020925 (2014). Rhodes, et al.: Sea ice as a source of sea salt aerosol to Greenland ice cores: a model-based study, Atmos. Chem. Phys., 17, 9417-9433, https://doi.org/10.5194/acp-17-9417-2017 (2017). Abbatt, et al: Halogen activation via interactions with environmental ice and snow in the polar lower troposphere and other regions, Atmos. Chem. Phys., 12, 6237-6271, https://doi.org/10.5194/acp-12-6237-2012, 2012.

Authors - The authors thank the reviewer for this insightful comment and have implemented it in the text. The text at the end of the introduction now reads: "Proposed mechanisms for enhanced sea salt aerosol concentrations include sublimating saline snow hydrometeors which has is hypothesized to produce sea salt aerosol at an order of magnitude higher production rate than over open ocean (Yang et al., 2008). This 'blowing snow' mechanism has been implemented in global models and various implementations have seen varying success in simulating sea salt concentrations (Levine et al., 2014; Huang and Jaeglé, 2016; Rhodes et al., 2017). However, there is the lack of experimental studies investigating the proposed mechanism of sea salt aerosol production from blowing snow, and the experimentally measured aerosol production rates. The results presented here, which vary in temporal resolution from low (days) to very high (minutes), suggest that coastal Antarctic aerosol composition cannot be explained without taking local meteorology (e.g. wind speed and direction) into account. Though the results presented here do not identify specific mechanisms, they strongly suggest that blowing snow may drive the composition dependence and aerosol enhancement observed as a function of wind speed, and further studies are necessary to investigate both production rates and mechanisms."

Reviewer - Section 3.7 (about bromine): Sander et al.'s (2003) global sea spray dataset clearly shows a general bromine depletion in micron mode, however, in ultra-fine mode, bromine is largely enriched. A similar phenomenon is also seen in polar dataset (Legrand et al., 2016). Do you have any comments on this size-dependent behaviour and relevance to your data interpretation? Sander, et al.: Inorganic bromine compounds in the marine boundary layer: A critical review, Atmos. Chem. Phys., 3, 1301–1336, 2003.

Authors - The authors thank the reviewer for bringing up these points. The following has been added to the text: "As a bulk measurement, mass fractions in the filter measurements are strongly biased towards larger, supermicron particles, whereas the AMS measurements are only sensitive to submicron particles. The below LOD concentrations of Br- in the filters combined with measurable Br with the AMS is generally consistent with measurements of the cycling of inorganic bromine in the marine boundary layer showing depletions of bromine in supermicron aerosol and enrichments in submicron aerosol (Sander et al., 2003). These results are also consistent with measurements at another coastal site in Antarctica (Legrand et al., 2016). This apparent size dependence for bromine in the aerosol phase supports models such as Legrand et al.'s (2016) that implement size-dependent depletion factors for bromine from aerosols. Additionally, the AMS measured Br as a function of wind speed results support the hypothesis that blowing snow provides a source of bromine. The local maxima of submicron Br concentrations above the wind speed threshold for blowing snow may suggest that gas-phase bromine is being liberated and absorbing or heterogeneously reacting onto the submicron aerosol population. Unfortunately, the lack of concurrent gas-phase bromine measurements means these results are not conclusive evidence for these hypotheses but do point out the need for more measurements."

Specific Comments Reviewer - P5L26: 'An' should be 'an'

Authors - This has been corrected.

Reviewer - P6L1-2: '50m' or '50mL'? A full stop is needed before the second '50 mL' (?)

Authors - The sentence has been changed to read: "The snow samples were collected when there was adequate snow in 50mL polyethylene plastic centrifuge tubes for future ion chromatography analysis"

Reviewer - P7L28: delete the duplicated word 'in'

Authors - This has been corrected.

Reviewer - SI Fig.S3: remove duplicated brackets in the caption.

Authors - This has been corrected.

Reviewer - SI Fig. S4: where are 'squares'? which 'solid' line (red or coloured) do you refer to?

Authors - The caption has been corrected to more accurately reflect the figure.

---

## Author Response (AR1)

**Response to Reviewer #1:**

The authors thank Reviewer #1 for their thoughtful and constructive comments on the manuscript. The reviewer's comments have been implemented as below:

**General Comments:**

There seems to be a shift of focus between the title/introduction and conclusions. While the manuscript puts emphasis on the influence of wind speed on the aerosol population the conclusion mainly focuses on the halogen cycle. A section in the introduction on the halogen cycle is needed. For example, the two last result subsections use the reference Legrand et al. (2017) heavily, while he is not mentioned in the introduction.

A section on the halogen cycle as it relates to Antarctic aerosols has been added to the introduction:

"Halogen-containing aerosols in the Antarctic are connected to the depletion of near-surface tropospheric ozone, a greenhouse gas, (Kalnajs et al., 2013) and may also contribution to new particle formation (Sipilä et al., 2016). Numerous remote sensing measurements of halogen oxides in the boundary layer (Saiz-Lopez et al., 2007; Simpson et al., 2015; Prados-Roman et al., 2018) show active bromine and iodine chemistry in the Antarctic troposphere. Chemical ionization mass spectrometry measurements of Br2, BrCl, and BrO confirmed the coupling of bromine and chlorine chemistry in the springtime Antarctic troposphere (Buys et al., 2013). Reactions with non-methane hydrocarbons (e.g. Ramacher et al., 1999), elemental mercury (Saiz-Lopez et al., 2008), and DMS (von Glasow et al., 2004; Read et al., 2008) can all have large impacts on tropospheric composition over the poles. Additionally, some work has also been done examining iodine oxides' role in forming ultrafine aerosols (O'Dowd et al., 1998; McFiggans et al., 2004; Saunders and Plane 2005). Despite significant progress in our understanding of atmospheric halogen gases, a lack of observational data capturing the interactions between gas-phase halogens, aerosols, and snow persists, limiting our understanding of these multiphase processes. "

The title is cryptic: "more than Source-Dependent Composition" is not a description of this work's content that will inform the reader directly of what this manuscript is about. What about: The importance of blowing snow to aerosol and the halogen cycle in coastal Antarctica: Influence of Source region versus wind speed.

The authors agree with the reviewer and have changed the title to incorporate the suggestion of the reviewer: new title "The importance of blowing snow to halogen-containing aerosol in coastal Antarctica: Influence of source region versus wind speed".

The notation of the chemical species throughout the manuscript is confusing: e.g., chloride, Cl, Cl-, Chl. I suggest to introduce a notation that makes clear with which instrument the species was measured, e.g., Cl- for IC results, Chl for AMS results.

Please check the manuscript throughout, sometimes the ionic charge is not provided (forgotten). Below are some hints, but I might not have spotted all. The authors thank the reviewer for pointing out that the nomenclature is not 100% consistent. A sentence has been added at the end of Section 3 denoting which notation implies which measurement. A few parenthetical reminders have also been added throughout the rest of the text due to the length of the manuscript.

"In this manuscript, 3 notations are presented for chemical identities and correspond to either instrument used to quantify or to their presence regardless of molecular structure. 1) For any ion measured with the IC system, ionic charge is noted (e.g. Cl-, Na+). 2) For compounds measured with the AMS, notation in the text follows AMS standard notation (e.g. Chl for chloride, SO4 for sulfate, NO3 for nitrate) with the exception of sodium measured by the AMS and corrected through the process described in S2.3 – NaAMS. 3) For general and/or hypothetical discussions without implication to measurement or molecular compounding, elements are referred to simply by their periodic table notation or written out (e.g. chloride, Cl, sodium, Na)."

Some exceptions are left in the text (e.g. "chloride salts" in S2.3) to avoid having to add another notation (e.g. XCI) but the meaning in these few cases should be obvious for all readers.

**Specific Comments**

Aerosol size distribution information based on AMS measurements are not provided in this manuscript. Therefore the description of size calibrations could be omitted to make the manuscript shorter.

The sentences describing the AMS calibrations have been replaced with: "The AMS was calibrated as described in Giordano et al., 2017"

In section 2.3, p. 7, second paragraph: Please include a brief description of the model by Salcedo et al. (2010) so the reader does not have to look up another publication to understand how the authors arrived from the e-folding time of 40 to the modelled Chl and Na concentrations as shown in Fig. 1. p. 9, l. 21: It is not clear whether those data points are shown as well. Please make this evident in the text.

A brief description of the Salcedo model has been added to the text:

"The model employed by Salcedo et al. (2010) assumes an incoming mass flux of a slow-vaporizing species which impacts the vaporizer, sticks to it, and then desorbs at a rate proportional to the amount on the vaporizer at any given time. This assumption results in an explicit ordinary differential equation."

A figure showing the data fit to obtain the e-folding time has been added to the supplement for this work (SI Fig. 2).

**p. 9, l. 27: "chloride losses" Do they refer as relative to Na+ or to the total chloride concentration?**

The text has been changed to reflect the cited manuscript: "Virkkula et al. (2006) noted chloride losses (relative to Na+) linked with..."

**p.* 10., *l.* 3: How can winter blizzard conditions be more consistent with the 20DIAC campaign which happened later in the season?**

The text states "blizzard and strong wind conditions", where "strong wind conditions" is consistent with what was measured in the Spring (2015) campaign (over 60% of the total time of that deployment seeing winds over 8ms-1). "Blizzard" is left in because that is part of how Hara et al., 2004 describes their results.

**p. 12, l. 30: Are those the total numbers of trajectories or just at one release height? Please specify in the manuscript.**

A parenthetical addition has been made to make the sentence clearer: "(37 and 36 days, respectively, at 3 hour intervals for each release height)".

**Figure 5: It would be helpful to make all data points grey and overlay them by wind speed binned averages with standard deviations.**

The figure has been modified as recommended.

**p. 14, l. 15f: What was the size range of particles that Hara et al. (2014) measured? From the text it is not evident that the comparison makes sense.**

"primarily in the 500nm -  $2\mu$ m size ranges" has been added to the text. This is on the upper range of what would be captured by the AMS but should be well encompassed by the Lighthouse OPC used.

**Section 3.7: Could the vertical displacement of the back trajectories be an indicator of possible processes? The authors looked at source regions which do not provide a clear hint, but potential influence of high tropospheric or stratospheric air masses could be important.**

The HYSPLIT trajectories examined often showed a lot of movement altitudinally over their lifetimes. However, analyzing how the vertical mobility of an air mass over its lifetime affects the halogen and/or aerosol content is outside of the scope of this manuscript. However, the reviewer brings up an excellent point. The following has been added to the text in S3.7 to make the reviewer's point known and to point out potential future work:

"The results presented here suggest that source regions for air masses do not significantly affect aerosol halogen concentrations in Antarctica. However, we do not here assess what, if any, impact that vertical displacement during transport may have on the halogen concentrations in the aerosol phase. Due to the low concentrations of aerosol-phase halogens measured, the uncertainty in the HYSPLIT runs, and the fact that the air masses examined in most of the HYSPLIT runs had large altitude changes during their back trajectory lifetimes, assessing the effect of vertical displacement of an air mass on halogen or aerosol concentrations or compositions is outside of the scope of this manuscript. However, it should be noted that models have shown there to be significant differences in the vertical distribution of gasphase halogens in the boundary layer (Lehrer et al., 2004; Saiz-Lopez et al., 2008). Any vertical movement of an air mass could therefore have significant impacts on the halogen cycle and halogenaerosol interactions." Conclusions: p. 22, l. 8: The mixing state of particles has not been mentioned before in the manuscript. If there is a strong argument for external halogen mixture, this needs to be included in the respective results section. It will also be highly informative to include a couple of sentences what more information of the halogen cycle in Antarctica will be important for.

The statement about mixing state has been removed since no evidence for it has been presented in the text. The following has been added to demonstrate the importance of understanding the halogen cycle: "A clearer understanding of the halogen cycle over Antarctica will result in better modelling capabilities of the oxidative capacity of the polar troposphere which impacts both current climate (e.g. DMS oxidation) and paleoclimate reconstructive abilities."

**Technical Comments:**

**(Grammatical/Spelling comments:)**

- p. 5, l. 26: "an" instead of "An"
- p. 7, l. 28: "and sodium my occur in the data."
- p. 8, l. 19: delete "while in the field and subsequent data analysis"

p. 9, l. 6: "it"

- p. 9, l. 27: "was" instead of "is not clear."
- p. 12, l. 15: "differences that"
- p. 13, l. 19: "Two" instead of "Several"
- p. 13, l. 25: "which in turn is a function: : :"
- p. 14, l. 14: "These results are: : :"
- p. 16, l. 9f and elsewhere: The hyphens turn out very long and spaces are missing.
- p. 17, l. 2: "region" instead of "point". The sources are rather regions that specific points.
- p. 17, l. 5: Delete "it was quickly noted that"
- p. 18, l. 5: " that the increased chloride concentrations as measured by the AMS are not: : :"
- p. 19, l. 13 and elsewhere: if this is the AMS derived ratio use Chl:Na
- p. 20, l. 14: "with the AMS" instead of "in the AMS"
- p. 20, l. 20: "by Maffezzoli" according to the sentence structure
- p. 21, l. 21: "summer", l. 26: delete "overall"
- p. 21, l. 29: Based on AMS measurements

The above grammatical/spelling comments have all been implemented into the manuscript.

**p. 17, l. 33 and elsewhere: sometimes it is written Figure x or Fig. x or figure. Check the journal style.**

All instances of "Figure" have been changed to "Fig." unless they are at the start of a sentence for consistency.

**p. 17, l. 16: replace "in the AMS" by "as measured by the AMS". The original formulation, here and elsewhere, sound like the species are generated within the AMS.**

There are a few instances where "in the AMS" is the correct usage in context (e.g. S2.3, other occurrences regarding vaporization) but the rest of the occurrences have been changed as suggested.

**p. 3, l. 10: elaborate which elements are meant by "marine elements"**

"(Li, K, Mg, Ca, Sr)" has been added to the text to represent what was measured (minus Na since it is a focus here) by Weller et al., 2008.

**p. 7, l. 11: "Figure 1 shows how the wind direction and: : :" The wind direction is not shown, but can be added easily.**

"...direction and..." referred to an older version of the figure and has been removed. Wind direction changes are not relevant to the decay period shown here.

**p. 8, l. 1: "freezer in Antarctica." Admitted, there is a limited total number of freezers in Antarctica, but more specificity is desirable, e.g., the research site's freezer.**

"in the Crary Lab" has been added to the text.

**p. 8, l. 21f: It is not clear what is meant by "increasing temporal resolution patterns."**

The sentence has been removed because it does not add anything to the context of the rest of the paragraph.

**p. 10, l. 24 / p. 11, l. 10-12: Do you mean "SO4 2-"?**

The authors thank the reviewer for this catch; the instances on p.10 and p.11 have been corrected to "2-"

**p. 11, l. 2f: The abbreviations have not been introduced.**

The abbreviations have been removed and the sentence changed to read:

"This is not in agreement with results from Eom et al. (2016) which measured higher inorganic salt concentrations in summer aerosols by Raman microspectrometry and attenuated total reflection Fourier transform infrared imaging techniques"

**p. 11, l. 32: sulphate or sulfate?**

"Sulphate" has been changed to "sulfate" to be consistent with the rest of the manuscript and abide by IUPAC recommendations.

**p. 12, l. 29: SI or Supplemental Information?**

"Supplemental Information" has been changed to "SI" for consistency throughout the manuscript.

**p. 14, l. 33: It is unclear what is meant by "make up the aerosol to inland continental snow."**

The sentence has been changed to make the meaning clearer:

"...may explain longer range transport of chemical species (found primarily in the aerosol phase) to inland continental snow."

**p. 15, l. 27: It is unclear what is meant by " by extension concentrations."**

The sentence has been changed to make the meaning clearer:

"...and, by extension, aerosol concentrations."

**p. 18, l. 13 and elsewhere: Do you refer to the IC measured species CI- and Na+? Or the AMS derived species Chl and Na?**

This comment is addressed in the "General Comments" section above.

**p. 19, l. 25: Do you mean at intermediate high wind speeds?**

"than at high wind speeds" has been removed to make the sentence clearer.

Sentence now reads: "This is near the same wind speed at which snow particle numbers increase more rapidly (7-13ms-1; Fig. 4)."

**p. 21, l. 27f: the notation of the chemical species is not clear,**

This comment is addressed in the "General Comments" section above.

**Response to Reviewer #2**

**General Comments**

The first part of the Introduction section contains an overview of Antarctic aerosol studies, but mainly in instrumental aspect. Given the title/research focus of this manuscript is about blowing snow and relevant aerosol, a brief introduction is needed. For example, on page 3 lines 27-29, it reads 'Some studies in Antarctica have hypothesized that elevated wind speeds can cause sea salt concentration differences through unknown mechanisms', then which studies do you refer to? References should be supplied here. Yang et al. (2008) has proposed a mechanism of SSA production from blowing snow through sublimating saline wind-blown snow particles, which parameterisation has been implemented in global models to investigate high latitude SSA (e.g. Levine et al., 2014; Huang and Jaegle 2016; Rhode et al., 2017). Frost flowers are also thought to be a SSA source. More information can be found in e.g. Abbatt et al., (2012).

Yang, et al.: Sea salt aerosol production and bromine release: Role of snow on sea ice, Geophys. Res. Lett., 35 (L16815), doi:10.1029/2008gl034536, 2008.

Huang J., Jaeglé, L., Wintertime enhancements of sea salt aerosol in polar regions consistent with a sea-ice source from blowing snow, ACP, doi:10.5194/acp-2016-972 (2016). Levine, et al.: Sea salt as an ice core proxy for past sea ice extent: A process-based model study, J. Geophys. Res. Atmos., 119, 5737–5756, doi:10.1002/2013JD020925 (2014). Rhodes, et al.: Sea ice as a source of sea salt aerosol to Greenland ice cores: a model-based study, Atmos. Chem. Phys., 17, 9417-9433, https://doi.org/10.5194/acp-17-9417-2017 (2017). Abbatt, et al: Halogen activation via interactions with environmental ice and snow in the polar lower troposphere and other regions, Atmos. Chem. Phys., 12, 6237-6271, https://doi.org/10.5194/acp-12-6237-2012, 2012.

The authors thank the reviewer for this insightful comment and have implemented it in the text. The text at the end of the introduction now reads:

"Proposed mechanisms for enhanced sea salt aerosol concentrations include sublimating saline snow hydrometeors which has is hypothesized to produce sea salt aerosol at an order of magnitude higher production rate than over open ocean (Yang et al., 2008). This 'blowing snow' mechanism has been implemented in global models and various implementations have seen varying success in simulating sea salt concentrations (Levine et al., 2014; Huang and Jaeglé, 2016; Rhodes et al., 2017). However, there is the lack of experimental studies investigating the proposed mechanism of sea salt aerosol production from blowing snow, and the experimentally measured aerosol production rates. The results presented here, which vary in temporal resolution from low (days) to very high (minutes), suggest that coastal Antarctic aerosol composition cannot be explained without taking local meteorology (e.g. wind speed and direction) into account. Though the results presented here do not identify specific mechanisms, they strongly suggest that blowing snow may drive the composition dependence and aerosol enhancement observed as a function of wind speed, and further studies are necessary to investigate both production rates and mechanisms." Section 3.7 (about bromine): Sander et al.'s (2003) global sea spray dataset clearly shows a general bromine depletion in micron mode, however, in ultra-fine mode, bromine is largely enriched. A similar phenomenon is also seen in polar dataset (Legrand et al., 2016). Do you have any comments on this size-dependent behaviour and relevance to your data interpretation?

Sander, et al.: Inorganic bromine compounds in the marine boundary layer: A critical review, *Atmos. Chem. Phys.*, 3, 1301–1336, 2003.

The authors thank the reviewer for bringing up these points. The following has been added to the text:

"As a bulk measurement, mass fractions in the filter measurements are strongly biased towards larger, supermicron particles, whereas the AMS measurements are only sensitive to submicron particles. The below LOD concentrations of Br- in the filters combined with measurable Br with the AMS is generally consistent with measurements of the cycling of inorganic bromine in the marine boundary layer showing depletions of bromine in supermicron aerosol and enrichments in submicron aerosol (Sander et al., 2003). These results are also consistent with measurements at another coastal site in Antarctica (Legrand et al., 2016). This apparent size dependence for bromine in the aerosol phase supports models such as Legrand et al.'s (2016) that implement size-dependent depletion factors for bromine from aerosols. Additionally, the AMS measured Br as a function of wind speed results support the hypothesis that blowing snow provides a source of bromine. The local maxima of submicron Br concentrations above the wind speed threshold for blowing snow may suggest that gas-phase bromine is being liberated and absorbing or heterogeneously reacting onto the submicron aerosol population. Unfortunately, the lack of concurrent gas-phase bromine measurements means these results are not conclusive evidence for these hypotheses but do point out the need for more measurements."

**Specific Comments**

**P5L26: 'An' should be 'an'** This has been corrected.

**P6L1-2: '50m' or '50mL'? A full stop is needed before the second '50 mL' (?)**

The sentence has been changed to read: "The snow samples were collected when there was adequate snow in 50mL polyethylene plastic centrifuge tubes for future ion chromatography analysis"

**P7L28: delete the duplicated word 'in'**

This has been corrected.

**SI Fig.S3: remove duplicated brackets in the caption.**

This has been corrected.

**SI Fig. S4: where are 'squares'? which 'solid' line (red or coloured) do you refer to?**

The caption has been corrected to more accurately reflect the figure.

**The importance of blowing snow to halogen-containing aerosol in coastal Antarctica: Influence of source region versus wind speed**

Michael R. Giordano1, Lars E. Kalnajs3, J. Douglas Goetz1,a, Anita M. Avery1,b, Erin Katz2, Nathaniel W. May4, Anna Leemon4, Claire Mattson4, Kerri A. Pratt4, Peter F. DeCarlo1,2

[revised manuscript text omitted]

---

## Editor Decision (ED1)

Review by Editor

I would like to thank the authors for comprehensively addressing all points raised by the Reviewers. I have added this additional, minor review, to pick up a few remaining typos, mis-labelling issues, and a few areas where additional clarification would help the reader.

These comments are not in order of priority, but roughly go in order through the paper.

Page 3 line 24: contribute, not contribution

Page 3 line 29, I suggest citing also e.g. Ariya et al., Tellus 56B, 397-403, 2004, which is specifically a mercury paper

Page 3 line 31: I suggest including reference to    Sipilä M, et al. (2016) Molecular-scale evidence of aerosol particle formation via sequential addition of HIO3. Nature 537:532–534, which also has some polar results, as opposed to the other references cited, which are mid-latitude.

Page 4 line 4: O'Dowd, not O'dowd

Page 4 line 9: "…hydrometeors which is hypothesized.." i.e take out the "has"

Page 5: in the description of the SP-AMS, please give a size range for the particles measured. Later in the paper (e.g. p21) there is discussion re sub-micron, but I cannot find mention of the size range captured by the instrument anywhere. Also please state whether the AMS is size-segregated or not (I believe it is not, but it would help the reader to clarify).

Page 10 line 26: normally mirabilite is referred to as $Na_2SO_4.10H_2O$ (not "X $10H_2O$") – please amend

Page 12 line 8: "falling on both sides…", i.e. remove the "and on"

Figure 2 – please sort out the caption and legend and make them consistent e.g. there are no red symbols…; spring/summer 2014 are gold, not blue; winter/spring 2015 are blue... Also, please state in the caption that winter/spring is 2015 to keep things consistent.

Page 14 line 8: HYSPLIT is Fig S4 not S3.

Page 16 line 15 to 16: the statement "From Fig 6 it is clear that there are no statistically significant differences in Chl:Na$_{AMS}$, neither inter- nor intra-seasonally" should be backed up by some numbers – please give the ratios and explain what basis you use to define statistically significant differences. Further, Spring 2015 Primary continental ratio seems very different to all the others – indeed in the opposite direction – and actually a careful look suggests that there are differences between other ratios as well (e.g. 2015 primary marine vs 2014 primary marine, or vs 2014 mixed, etc…). Additional clarification to support the statement made is needed here.

Page 16 line 17 and 18: As above, the authors need to provide additional information to support their statement that "Related to this is that there are no intra-seasonal differences in any of the other aerosol species, with the exception of summer (2014) continental sulfate concentrations compared to marine or mixed concentrations". Again, please provide numbers with which this

statement can be supported, and explain the basis of the statement (i.e. how do you define a "difference"). Again, a careful look suggests there are a number of differences, including for sulfate in 2015, and also for organics. Please review/clarify.

Page 18 line 8: please add "within a season" to the statement about chloride increasing at high winds only, i.e. "first, that within a season, the absolute amount of chloride…" – because e.g. low wind in 2015 had more chloride than high wind SE in 2014…

Page 18 line 10, referring to Fig S6 – Ammonium in summer 2014 almost seems to be anti-correlated with wind speed… Suggest some refinement to wording.

Page 18 line 23 – Section 3.1.3 should presumably say Section 3.1.2

Page 19 line 16: presumably the authors mean Fig S6?

Page 20 line 4: given the comments above, I'd suggest changing the first sentence in Section 3.6 to read: "The previous section demonstrated that, for a given season, overall mass of chloride and sodium are dependent…"

Fig S8, please add in the caption what are the solid green and dashed blue lines (presumably sea water, and seawater mirabilite-depleted, respectively) – this would help the reader.

Fig 7 and Fig S8 – please remind the reader that NW = marine and SE = continental air

Page 21 line 16: the Maffezzoli et al (2017) reference is irrelevant of the discussion here if they did not show iodide (and can therefore not contribute to the discussion here about bromide to iodide ratios); please remove the reference.

Conclusions – please review in light of above suggested changes to make sure everything is consistent.

Fig 4 – please state in the figure caption that the Laser-disdrometer measurements were made at 5m above the snow surface.

---

## Author Response (AR2)

Editor:

Page 3 line 24: contribute, not contribution

Page 3 line 29, I suggest citing also e.g. Ariya et al., Tellus 56B, 397-403, 2004, which is specifically a mercury paper

Page 3 line 31: I suggest including reference to Sipilä M, et al. (2016) Molecular-scale evidence of aerosol particle formation via sequential addition of HIO3. Nature 537:532–534, which also has some polar results, as opposed to the other references cited, which are mid-latitude.

Page 4 line 4: O'Dowd, not O'dowd

Page 4 line 9: "…hydrometeors which is hypothesized.." i.e take out the "has"

Page 10 line 26: normally mirabilite is referred to as Na2SO4.10H2O (not "X 10H2O") – please amend

Page 12 line 8: "falling on both sides…", i.e. remove the "and on"

Figure 2 – please sort out the caption and legend and make them consistent e.g. there are no red symbols…; spring/summer 2014 are gold, not blue; winter/spring 2015 are blue... Also, please state in the caption that winter/spring is 2015 to keep things consistent.

Page 14 line 8: HYSPLIT is Fig S4 not S3.

Page 18 line 8: please add "within a season" to the statement about chloride increasing at high winds only, i.e. "first, that within a season, the absolute amount of chloride…" – because e.g. low wind in 2015 had more chloride than high wind SE in 2014…

Page 18 line 23 – Section 3.1.3 should presumably say Section 3.1.2

Page 20 line 4: given the comments above, I'd suggest changing the first sentence in Section 3.6 to read: "The previous section demonstrated that, for a given season, overall mass of chloride and sodium are dependent…"

Fig S8, please add in the caption what are the solid green and dashed blue lines (presumably sea water, and seawater mirabilite-depleted, respectively) – this would help the reader.

Fig 7 and Fig S8 – please remind the reader that NW = marine and SE = continental air

Page 21 line 16: the Maffezzoli et al (2017) reference is irrelevant of the discussion here if they did not show iodide (and can therefore not contribute to the discussion here about bromide to iodide ratios); please remove the reference.

Fig 4 – please state in the figure caption that the Laser-disdrometer measurements were made at 5m above the snow surface.

Authors Response:

The above changes have all been made and are highlighted in blue (or deleted where appropriate).

Editor:

Page 5: in the description of the SP-AMS, please give a size range for the particles measured. Later in the paper (e.g. p21) there is discussion re sub-micron, but I cannot find mention of the size range captured by the instrument anywhere. Also please state whether the AMS is size-segregated or not (I believe it is not, but it would help the reader to clarify).

Authors:

The        following        has        been        added        to        the        text:

…In both field seasons aerosol composition was measured by a Soot Particle Aerosol Mass Spectrometer (SP-AMS; Aerodyne Research Inc., Billerica MA; Onasch et al., 2012). The SP-AMS consists of an Aerodyne High-Resolution Time-of-Flight

5    Aerosol Mass Spectrometer (HR-ToF-AMS; DeCarlo et al., 2006) combined with a soot vaporizing laser (from Droplet Measurement Tech., Boulder CO), though the soot laser was not used during 2ODIAC. Transmission efficiencies for different instruments vary, but generally particles between 60nm - 600nm vacuum aerodynamic diameter are transmitted with unit efficiency, with fraction of transmitted particle decreasing for larger sizes (Liu et al. 2007).  Only particle bulk composition is discussed here due to the lower signal/noise associated with particle sizing. …

Editor:

Page 16 line 15 to 16: the statement "From Fig 6 it is clear that there are no statistically significant differences in Chl:NaAMS, neither inter- nor intra-seasonally" should be backed up by some numbers – please give the ratios and explain what basis you use to define statistically significant differences. Further, Spring 2015 Primary continental ratio seems very different to all the

15    others – indeed in the opposite direction – and actually a careful look suggests that there are differences between other ratios as well (e.g. 2015 primary marine vs 2014 primary marine, or vs 2014 mixed, etc…). Additional clarification to support the statement made is needed here.

and

Page 16 line 17 and 18: As above, the authors need to provide additional information to support their statement that "Related

20    to this is that there are no intra-seasonal differences in any of the other aerosol species, with the exception of summer (2014) continental sulfate concentrations compared to marine or mixed concentrations". Again, please provide numbers with which this statement can be supported, and explain the basis of the statement (i.e. how do you define a "difference"). Again, a careful look suggests there are a number of differences, including for sulfate in 2015, and also for organics. Please review/clarify.

25    Authors:

The following has been added to the text:

    . From Fig. 6 it is clear that there are no statistically significant differences in Chl:$Na_{AMS}$ intra-seasonally, i.e. between different air masses for a given measurement campaign season (average Chl:$Na_{AMS}$ = 1.9 ± 0.3 for the 2014 spring/summer and 1.0 ± 0.1 for the 2015 winter/spring measurements; all values shown in Table S1). It is unclear why the continental air

30    masses measured in the 2015 winter/spring season are higher than the marine and mixed air masses but the Chl:$Na_{AMS}$ ratio is consistent within measurement and propagated error. The absolute concentrations of both ammonium and nitrate also remain consistent within measurement error across the various air mass types for a given campaign season. This is inconsistent with previous measurements over similar size distributions by impactors by Virkkula et al. (2006). Organics concentrations do show small variations in the summer marine air masses (10.4 ± 1.4 ng m$^{-3}$) as compared to the other summer air masses (2.6 ± 0.8

and 2.1 ± 0.5 ng m$^{-3}$ continental and mixed, respectively) which is unsurprising considering summer oceanic phytoplankton activity. Conversely, lower organics concentrations in the marine spring air masses were also observed (0.2 ± 2.5 as compared to 10.1 ± 2.0 and 4.7 ± 2.2 ng m$^{-3}$ for the continental and mixed air masses, respectively). It is unclear why organics are so much higher in the continental spring air masses but may be related to sustained high winds over the continent lofting and transporting previously deposited organics from the previous seasons. The higher organics may also simply be due to higher energy requirements for the human settlements on the continent in the winter. More work is needed, however, to determine if this is a sustained trend or not. Sulfate concentrations are also higher in the continental air masses for both seasons as compared to the marine and mixed air masses and are likely related to the results discussed in Giordano et al. (2017).

Editor:

Page 18 line 10, referring to Fig S6 – Ammonium in summer 2014 almost seems to be anti-correlated with wind speed… Suggest some refinement to wording.

Authors:

The section has been added to make the meaning clear:

Three major observations are apparent in Fig. 7 (and SI Fig. S6): first, that within a season, the absolute amount of chloride increases at high wind speeds only, with little variation between low and medium wind speeds; second, that the absolute amount of organics and sulfate increase with wind speed in the springtime measurements only; and third, that the absolute amounts of nitrate and ammonium do not statistically significantly respond to wind speed and are relatively independent of wind direction. For the 2014 spring/summer measurements, nitrate ranges from 1.6-2.6 ng m$^{-3}$ with a standard deviation of ±1 ng m$^{-3}$ and ammonium ranges from 5.0-9.0 ng m$^{-3}$ with a standard deviation of ±2.0 ng m$^{-3}$. For the 2015 winter/spring measurements, the values are 3.8-5.6 ± 2 ng m$^{-3}$ and 17-21 ± 3 ng m$^{-3}$ for nitrate and ammonium, respectively.

Editor:

Page 19 line 16: presumably the authors mean Fig S6?

Authors:

This has been changed to "Fig. 7 (and Fig. S6)"

Editor:

Conclusions – please review in light of above suggested changes to make sure everything is consistent.

Authors:

The conclusions deal mostly with the halogens discussed in the text and the above edits deal primarily with the non-halogen species so no changes are made. We do not believe we can draw any definitive conclusions on the non-halogen species' wind speed or air mass dependencies and include the discussion in the text primarily for the sake of completeness. We discuss in the main text how further work is needed before any such conclusions can be drawn for these species.

[revised manuscript text omitted]